# CrossGNN: Confronting Noisy Multivariate Time Series Via Cross Interaction Refinement

Qihe Huang[†,§], Lei Shen[§], Ruixin Zhang[§], Shouhong Ding[§], Binwu Wang[†],
Zhengyang Zhou[†,‡,✉], Yang Wang[†,‡,✉]

[†] University of Science and Technology of China (USTC), Hefei, China
[‡] Suzhou Institute for Advanced Research, USTC, Suzhou, China
[§] Youtu Laboratory, Tencent, Shanghai, China
Email: {hqh, wbw1995}@mail.ustc.edu.cn, {zzy0929, angyan}@ustc.edu.cn,
shenlei1996@gmail.com, {ruixinzhang, ericshding}@tencent.com

## Abstract

Recently, multivariate time series (MTS) forecasting techniques have seen rapid development and widespread applications across various fields. Transformer-based and GNN-based methods have shown promising potential due to their strong ability to model interaction of time and variables. However, by conducting a comprehensive analysis of the real-world data, we observe that the temporal fluctuations and heterogeneity between variables are not well handled by existing methods. To address the above issues, we propose CrossGNN, a linear complexity GNN model to refine the cross-scale and cross-variable interaction for MTS. To deal with the unexpected noise in time dimension, an adaptive multi-scale identifier (AMSI) is leveraged to construct multi-scale time series with reduced noise. A Cross-Scale GNN is proposed to extract the scales with clearer trend and weaker noise. Cross-Variable GNN is proposed to utilize the homogeneity and heterogeneity between different variables. By simultaneously focusing on edges with higher saliency scores and constraining those edges with lower scores, the time and space complexity (i.e., $O(L)$) of CrossGNN can be linear with the input sequence length $L$. Extensive experimental results on 8 real-world MTS datasets demonstrate the effectiveness of CrossGNN compared with state-of-the-art methods. The code is available at https://github.com/hqh0728/CrossGNN.

## 1 Introduction

Time series forecasting has been widely used in many fields (i.e., climate [1], traffic [31], energy [3], finance [13], etc) [8, 29, 20, 10]. Multivariate time series (MTS) consists of time series with multiple variables and MTS forecasting aims at predicting future values based on historical time series. Deep learning models [26, 24, 17, 4, 21] have demonstrated superior performance in MTS forecasting. In particular, Transformer-based models [34, 27, 35] have achieved great power in MTS, benefiting from its attention mechanism which can model the long-term interaction between different time points of sequences (cross-time). Graph Neural Networks (GNNs) [28, 16, 2, 19, 7] have also shown promising results for MTS forecasting, which can extract pre-defined or adaptive interaction between different variables (cross-variable).

However, a recent study [33] shows that a simple linear model dramatically outperformed many state-of-the-art (SOTA) models, and it inspires us to investigate the reasons why the existing cross-time

---

This work is done when Qihe Huang was an intern at YouTu Lab.
✉ Yang Wang and Zhengyang Zhou are corresponding authors.

37th Conference on Neural Information Processing Systems (NeurIPS 2023).

and cross-variable interaction modeling fail to enhance prediction performance. By conducting a thorough analysis on real-world data, we observe that the presence of some unexpected noise (caused by humans, sensor distortion) may be responsible for it. **In time dimension**, as shown in Figure 1 (a), Transformer-based model heavily relies on the input sequences to generate the attention map, while its prediction may be susceptible to incidental noise, even some small fluctuation (i.e., noise) could easily lead to significant shifts on temporal dependencies. Our findings reveal that self-attention mechanism tend to assign high scores to outlier points in time series, resulting in spurious cross-time correlations. **In variable dimension**, cross-variable correlation exhibits a complex and dynamic evolution over time [18]. Despite the existence of underlying causal associations between variables, extracting cross-variable interaction is difficult due to the influence of noise interference. Additionally, as shown in Figure1 (b), we observe that such unexpected noise, which can be detected by the outlier detection algorithm [14], accounts for a high proportion in the time series.

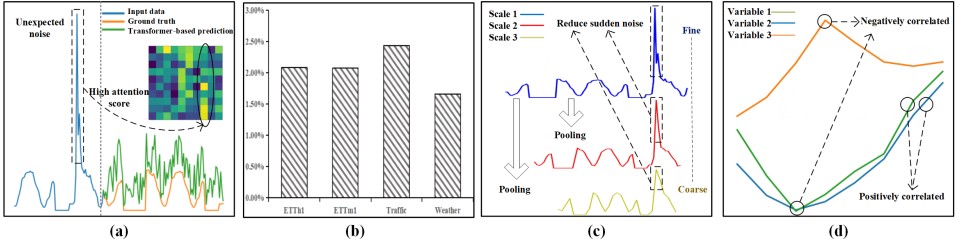

Figure 1: Data analysis on the real-world dataset [31]. (a) Forecasting results of Transformer-based models suffering from unexpected noise. (b) Proportion of noise in the ETTh1, ETTm1, Traffic, and Weather datasets [27] detected by [14]. (c) Different levels of noise signals in multi-scale time series. (d) Homogeneous and heterogeneous relationships between variables.

Despite the non-negligible noise in time series, we can still discover the potential opportunities to confront noise challenges. **(1) Cross-Scale Interaction.** As shown on Figure 1 (c), by performing multi-scale extraction on the time series, we observe that different scales possess distinct levels of noise intensity, typically with coarser scales exhibiting lower noise intensity. Evidently, capturing dependencies across different scales enables the cross-time relationships to be robust against noise [15]. **(2) Cross-Variable Interaction.** As shown on Figure 1 (d), there is both homogeneity and heterogeneity for cross-variable interaction in real-world data [36]. In fact, the two relationships can both contribute to invariant connections during the temporal progression. Consequently, learning the invariant associations that contain both homogeneous and heterogeneous relationships among variables can boost its robustness to confront noise. Based on above analysis, it is still challenging to refine interaction in noisy MTS. The key obstacles can be summarized as follows: 1) How to capture cross-scale interaction that is not sensitive to unexpected input noise. 2) How to extract cross-variable relations between heterogeneous variables.

In this work, we propose CrossGNN, which is the first GNN solution to refine both cross-time and cross-variable interactions for MTS forecasting. To deal with the unexpected noise in time dimension, we firstly devise an adaptive multi-scale identifier (AMSI) to construct multi-scale time series with different noise levels. In time dimension, we propose Cross-Scale GNN, which is a temporal correlation graph, to model the dependency between different scales. The scales with clearer trends and weaker noise will be assigned with more edge weight. In variable dimension, we first introduce heterogeneous interaction modeling between variables into MTS forecasting and propose cross-variable GNN to utilize the homogeneity and heterogeneity between different variables with positive and negative edge weights. By focusing on edges with higher saliency scores and constraining those edges with lower scores at the same time, CrossGNN achieves linear time and space complexity (i.e., $O(L)$) with input sequence length $L$. The main contributions are summarized as follows:

- We conduct comprehensive studies on real-world MTS data and discover that the unexpected noise in time dimension and variable-wise heterogeneity between variables is not well handled by existing Transformer-based and GNN-based models.

- We propose a linear complexity CrossGNN model, which is the first GNN model to refine both cross-scale and cross-variable interaction for MTS forecasting.

1) To deal with the unexpected noise in time dimension, AMSI is leveraged to construct multi-scale time series with different noise level and a Cross-Scale GNN is proposed to capture the scales with clearer trend and weaker noise.

2) Cross-Variable GNN is designed to model the dynamic correlations between different variables. This the first model to introduce heterogeneous interaction modeling between variables into MTS forecasting.

- Extensive evaluation on 8 real-world MTS datasets demonstrates the effectiveness of Cross-GNN. Specifically, CrossGNN achieved top-1 performance on 47 settings and top-2 on 9 settings when compared with 9 state-of-the-art models with varying prediction lengths.

## 2 Related Work

**Multivariate Time Series Forecasting.** MTS forecasting has witnessed significant advancements due to the emergence of deep neural networks. These networks can be based on Convolutional Neural Network (CNN) [26, 23], Recurrent Neural Network (RNN) [5, 6], Transformer [11, 25, 27, 35, 9], or Graph Neural Network (GNN) [28, 32]. Generally, the primary emphasis of these studies lies in devising interactions between the temporal dimensions (cross-time) and the variable dimensions (cross-variable).

**Cross-Time Interaction Modeling.** Cross-time interaction modeling aims to capture correlations between different time points. Recently, CNN-based model TimesNet [26] transforms the time series into a two-dimensional matrix and uses a CNN-based backbone for feature extraction. RNN-based model LSTnet [6] utilizes the Long Short-Term Memory (LSTM) to model the temporal dependencies, but it may be limited by the inherent issue of gradient vanishing/exploding in RNNs. Transformer-based models benefit from its self-attention mechanism, enabling them to capture long-term cross-time dependency. AutoFormer [27] incorporates a decomposition mechanism that splits the input sequence into trend and seasonality, and integrate an auto-correlation module into the transformer to capture long-term cross-time dependency. FedFormer [35] leverages a frequency-enhanced decomposition mechanism while incorporating additional frequency information. However, despite the outstanding performance of Transformer-based methods, we observe that their self-attention mechanism is susceptible to unexpected noise, as shown on Figure 1(a). Based on these findings, we propose an innovative GNN-based method that constructs a cross-scale temporal graph to mitigate the impact of temporal noise on modeling cross-time correlations.

**Cross-Variable Interaction Modeling.** Cross-Variable Interaction is proved to be critical for MTS forecasting [34], and numerous works have employed Graph Neural Networks (GNNs) [22, 37, 30, 38] to capture cross-variable relationships. STGCN [32] firstly utilizes GNN to model the cross-variable dependency in traffic forecasting, which can effectively capture the dependency between different roads in pre-defined topology graphs. MTGNN [28] expands the utilization of GNN from spatio-temporal prediction to MTS forecasting, and proposes a straightforward method for computing adaptive cross-variable graph. On the other hand, the Transformer-based MTS prediction works have also recognized the potential of cross-variable interactions to enhance prediction performance, such as CrossFormer [34]. However, the cross-variable relationship is dynamic and can be greatly influenced by noise during the learning process. Given this, we refine the cross-variable relationship by decoupling homogeneity and heterogeneity in MTS, resulting in a noise-insensitive relationship during the temporal evolution.

## 3 Methodology

In long-term multivariate time series (MTS) forecasting, the input comprises historical sequences across $D$ variables denoted by $\mathcal{X} = \{X_1^t, ..., X_D^t\}_{t=1}^L \in \mathbb{R}^{L \times D}$, where $L$ denotes the look-back window size and $X_i^t$ denotes time series of the $i_{th}$ variate at the $t_{th}$ time step. The objective of MTS forecasting is to predict future time series denoted by $\hat{\mathcal{X}} = \{\hat{X}_1^t, ..., \hat{X}_D^t\}_{t=L+1}^{L+T} \in \mathbb{R}^{T \times D}$ based on $\mathcal{X}$, where $T$ represents the prediction time steps and $T \gg 1$. The detailed structure of CrossGNN is illustrated in Figure 2. We firstly employ an adaptive multi-scale identifier (AMSI) to generate multi-scale time series and reduce noise on coarse scale. Then, we construct scale-sensitive and trend-aware temporal graph to extract cross-scale interaction. We perform cross-variable aggregation via

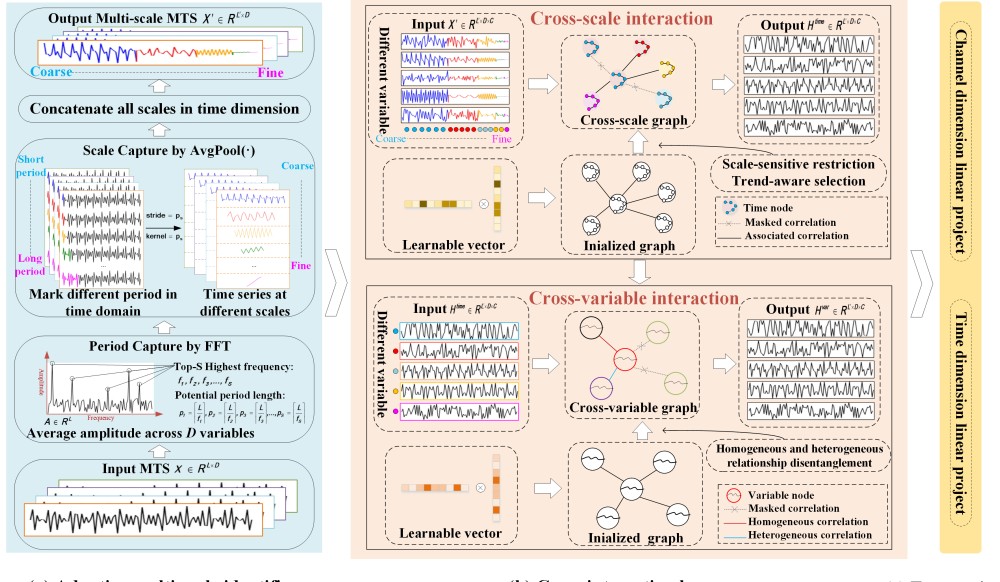

| (a) Adaptive multi scale identifier | (b) Cross interaction layer | (c) Forecasting |

Figure 2: CrossGNN architecture. (a) Adaptive multi-scale identifier (AMSI) is devised to extract the multi-scale MTS $\mathcal{X}'$ from the input $\mathcal{X}$. (b) Cross-Scale GNN facilitates cross-time interaction within and across different scales, while Cross-Variable GNN models cross-variable interaction for both homogeneous and heterogeneous relationships. (c) Direct multi-step (DMS) forecasting is leveraged to predict the future time series based on two MLPs.

modeling homogeneous and heterogeneous relationships between variables. Finally, direct multi-step (DMS) forecasting is adopted in decoder to predict future time series.

## 3.1 Adaptive Multi-Scale Identifier

The adaptive multi-scale identifier (AMSI) is designed to capture the different scales of MTS from coarse to fine, and reduce the unexpected noise on the coarse scale. Technically, we utilize Fast Fourier Transform (FFT) to analyze the time series in the frequency domain and calculate the potential periods of the time series, inspired by [26]. We compute the amplitude of each time series at various frequencies using FFT and subsequently average the amplitude of $\mathcal{X}$ across variable dimension:

$$A = \mathrm{Avg}_{i=1}^{D}\left(\mathrm{Amp}\left(\mathrm{FFT}\left(\mathcal{X}\right)\right)\right), \tag{1}$$

where $\mathrm{Amp}(\cdot)$ is the calculation of amplitude, $\mathrm{FFT}(\cdot)$ is the calculation of FFT, $A \in \mathbb{R}^L$ represents the calculated amplitude of each frequency, which is averaged from $D$ variables by $\mathrm{Avg}(\cdot)$. We choose the frequencies $\{f_1, f_2, \cdots, f_S\}$ which correspond to the Top-S amplitude values:

$$\{f_1, \cdots, f_S\} = \arg \mathrm{Top\text{-}S}(A), \tag{2}$$

where $\arg \mathrm{Top\text{-}S}(\cdot)$ picks out the $S$ frequency values with the highest amplitude from $A$. The period lengths $\{p_1, p_2, \cdots, p_S\}$ are calculated by the selected frequencies as follows:

$$p_s = \left\lceil \frac{L}{f_s} \right\rceil, s \in \{1, \cdots, S\}. \tag{3}$$

Then, $\mathrm{AvgPool}(\cdot)$ with kernel size $p_s$ and stride $p_s$, $s \in \{1, 2, ..., S\}$, are applied to the MTS $\mathcal{X} \in \mathbb{R}^{L \times D}$ in the time dimension to capture the MTS at $s$-th scales:

$$\mathcal{X}_s = \mathrm{AvgPool}(\mathcal{X})_{kernel=stride=p_s}, \tag{4}$$

where $\mathrm{AvgPool}(\cdot)$ downsamples time series to obtain $\mathcal{X}_s \in \mathbb{R}^{L(s) \times D}$, $L(s) = \lfloor \frac{L}{p_s} \rfloor$ is the time series length in the $s$-th scale, $\lfloor \cdot \rfloor$ is the operation of rounding down. We concatenate the captured different

scales in the time dimension and obtain a period-wise multi-scale MTS $\mathcal{X}' \in \mathbb{R}^{L' \times D}$ as the output of AMSI, $L' = \sum_{s=1}^{S} L(s)$ is the sum of lengths across all scales:

$$\mathcal{X}' = \text{Concat}(\mathcal{X}_1, \mathcal{X}_2, ..., \mathcal{X}_S). \tag{5}$$

Here, we employ an expansion dimension strategy (using an MLP), to create an embedding for each time step at different scale. This strategy is inherited from MTGNN [28], aiming to enhance the local semantics at each time step and positively impact subsequent cross-scale and cross-variable interactions. The shape of $X'$ is finally expanded as $\mathbb{R}^{L \times D \times C}$.

## 3.2 Cross-Scale GNN

The Cross-Scale GNN is designed to utilize the interaction of the multi-scale MTS $\mathcal{X}'$ and extract the scales with clearer association and weaker noise. The cross-scale graph in the time dimension is represented as $G^{scale} = (V^{scale}, E^{scale})$. $V^{scale} = \{v_1^{scale}, v_2^{scale}, ..., v_{L'}^{scale}\}$ is the time nodes set in all time scales, where $v_i^{scale}$ is the $i$-th time node. $E^{scale} \in \mathbb{R}^{L' \times L'}$ assigns the correlation weights between each time node, each element in $E^{scale}$ meaning the correlation weight between two time nodes (inter-scale or intra-scale). The key purpose of Cross-scale GNN is to learn the cross-scale temporal correlation weights $E^{scale}$ that is insensitive to noise interference. To diminish the effect of noise on correlation weights, we maintain the independence by initializing $E^{scale}$ with the production of two learnable vectors $vec_1^{scale}$ and $vec_2^{sclae}$, ensuring $E^{scale}$ not affected by noise hidden in input:

$$E^{scale} = \text{Softmax}(\text{ReLU}(vec_1^{scale} \times vec_2^{scale})), \tag{6}$$

where $\text{ReLU}(\cdot)$ is the active function regularizing the weight matrix so that each element is positive, and $\text{Softmax}(\cdot)$ is the operation to ensure the weights of all nodes correlated to a particular time node sum to 1.

**Scale-sensitive Restriction.** We consider that for any time node, the number of its correlated time nodes on the fine scale should be more than the number on the coarse scale. For any time node $v_i^{scale}$, the number of its correlated nodes at $s$-th scale is restricted to $k_s = \lceil \frac{K}{p_s} \rceil$, where $p_s$ is the period length of $s$-th scale, $\lceil \cdot \rceil$ denotes the ceiling function, $K$ is a constant. This ensures that finer-scale time series contribute more temporal node associations. The neighbor time node set at $s$-th scale (i.e. the correlated time nodes at $s$-th scale ) of $v_i^{scale}$ is denoted as:

$$\mathcal{N}_s^{scale}(v_i^{scale}) = \arg \text{Top-k}_s(E_s^{scale}(v_i^{scale})), \tag{7}$$

where $\arg \text{Top-k}_s(\cdot)$ is an operation to extract $k_s$ nodes with highest correlation weight, $E_s^{scale}(v_i^{scale}) \in \mathbb{R}^{L(s)}$ is the correlation weight of time node $v_i^{scale}$ at $s$-th scale. In this way, the number of neighboring nodes at different scales can be restricted based on the correlation weight matrix $E^{scale}$.

**Trend-aware Selection.** To ensure the temporal trends can be captured, we preserve the associations between a time node and its both preceding and succeeding nodes. Denote $\mathcal{N}^{trend}(v_i^{scale})$ as the trend neighbor set of time node $v_i^{scale}$, which can be defined as follows:

$$\mathcal{N}^{trend}(v_i^{scale}) = \{v_j^{scale} \mid |i - j| \leq 1, scale(v_i^{scale}) = scale(v_j^{scale})\}, \tag{8}$$

where $scale(\cdot)$ provides the scale of a time node. The trend neighbor set of $v_i^{scale}$ consists of its adjacent time nodes (i.e., $|i - j| \leq 1$) which share the same scale. This gives the ability to preserve temporal trends in the cross-scale correlation graph.

**Correlation Weight Re-normalization.** Denote $\mathcal{N}(v_i) = \mathcal{N}^{scale}(v_i) \cup \mathcal{N}^{trend}(v_i)$ as the selected neighbor set of time nodes $v_i^{scale}$. The final correlation weight is re-normalized as follows:

$$E^{scale}[i,j] = \begin{cases} \frac{E^{scale}[i,j]}{\sum_{v_m \in \mathcal{N}(v_i)} E^{scale}[i,m]}, & \text{if } v_j \in \mathcal{N}(v_i), \\ 0, & \text{otherwise,} \end{cases} \tag{9}$$

where $E^{scale}[i,j]$ is the correlation weight between $v_i^{scale}$ and $v_j^{scale}$. This step filters out non-significant correlation and preserves a restricted set of neighboring nodes $\mathcal{N}(v_i^{scale})$ to each node. Also, re-normalization is applied to the retained correlation. Then, cross-scale correlation graph between time nodes at different scales is constructed.

**Cross-scale interaction.** After obtaining the cross-scale temporal correlation graph, we do cross-scale interaction in the time dimension based on GNN, and the information propagation process will be stacked for $N$ layers:

$$\mathcal{HN}_{i,:}^{time,N} = (\sum_{s=1}^{S} \sum_{v_j \in \mathcal{N}_s^{scale}(v_i)} E[i,j] \cdot \mathcal{H}_{j,:}^{time,N-1}) + (\sum_{v_m \in \mathcal{N}^{trend}(v_i)} E[i,m] \cdot \mathcal{H}_{m,:}^{time,N-1})$$

$$\mathcal{H}_{i,:}^{time,N} = \sigma(Concat(\mathcal{HN}_{i,:}^{time,N}, \mathcal{H}_{i,:}^{time,N-1}) \cdot W)$$

$$\mathcal{H}_{i,:}^{time,N} = \mathcal{H}_{i,:}^{time,N}/||\mathcal{H}_{i,:}^{time,N}||_2$$

$$(10)$$

where $\sigma$ is the activation function; $W$ is the learnable matrix; $\mathcal{H}^{time}$ is the time node feature; $\mathcal{HN}^{time}$ is the aggregation of neighbor time node feature. $\mathcal{HN}_{i,:}^{time,N}$ aggregates the neighboring node feature from the previous layer correlated to the time nodes of $v_i^{scale}$. Then $\mathcal{H}_{i,:}^{time,N}$ is updated by the aggregated time node feature $\mathcal{HN}_{i,:}^{time,N}$ and its previous layer feature $\mathcal{H}_{i,:}^{time,N-1}$. Finally, the normalized $\mathcal{H}_{:,:}^{time,N}$ at $N$-th layer is the output of Cross-Scale GNN.

## 3.3 Cross-Variable GNN

Cross-Variable GNN is designed to extract the invariant correlation consisting of homogeneous and heterogeneous relationship. In variable dimension, the cross-variable graph is represented as $G^{var} = (V^{var}, E^{var})$. $V^{var} = \{v_1^{var}, v_2^{var}, ..., v_D^{var}\}$ is the variable node set, where $D$ is the number of variables, and $v_i^{var}$ is the $i$-th variable node. $E^{var}$ is the variable correlation weight matrix, each element in $E^{var}$ meaning the correlation weight between two variables. $E^{var}$ is initialized by production of two latent vectors $vec_1^{var}$ and $vec_2^{var}$:

$$E^{var} = \text{Softmax}(\text{ReLU}(vec_1^{var} \times vec_2^{var})) \tag{11}$$

**Heterogeneity Disentanglement.** Specifically, we select the nodes with $K_+^{var}$ highest correlation weight as positive neighbors with homogeneous connections, and take the nodes with $K_-^{var}$ lowest correlation score as negative neighbors with heterogeneous connections. Denote $E^{var}(v_i^{var})$ as the correlation weight of $D$ variable nodes related to $v_i^{var}$. For variable $v_i$, its two decoupled neighbor sets can be represented as $\mathcal{N}_-^{var}(v_i) = \text{Bottom-K}_-^{var}(E^{var}(v_i))$ and $\mathcal{N}_+^{var}(v_i) = \text{Top-K}_+^{var}(E^{var}(v_i))$, respectively.

**Correlation Weight Re-normalization.** The corresponding homogeneous and heterogeneous correlation weights are derived as follows:

$$E^{var}[i,j] = \begin{cases} -\frac{\frac{1}{E^{var}[i,j]}}{\sum_{v_k \in \mathcal{N}_-^{var}(v_i)} \frac{1}{E^{var}[i,k]}}, & \text{if } v_j \in \mathcal{N}_-^{var}(v_i), \\ \frac{E^{var}[i,j]}{\sum_{v_k \in \mathcal{N}_+^{var}(v_i)} E^{var}[i,k]}, & \text{if } v_j \in \mathcal{N}_+^{var}(v_i), \\ 0, & \text{otherwise} \end{cases} \tag{12}$$

This process filters out edges other than homogeneous and heterogeneous edges. The weights of homogeneous edges are positively correlated with their correlation scores, while the weights of heterogeneous edges are negatively correlated with their correlation scores. Additionally, separate re-normalization is applied to the weights of homogeneous edges and heterogeneous edges, respectively. Then, the cross-variable graph is constructed with disentangled homogeneous and heterogeneous correlations.

**Cross-variable Interaction.** For variable $v_i$, the disentangled cross-variable message passing can be formulated as:

$$\mathcal{HN}_{:,i}^{var,N} = \sum_{v_j \in \mathcal{N}_+^{var}(v_i)} E^{var}[i,j] \cdot \mathcal{H}_{:,j}^{var,N-1} + \sum_{v_k \in \mathcal{N}_-^{var}(v_i)} E^{var}[i,k] \cdot \mathcal{H}_{:,k}^{var,N-1},$$

$$\mathcal{H}_{:,i}^{var,N} = \sigma(Concat(\mathcal{HN}_{:,i}^{var,N}, \mathcal{H}_{:,i}^{var,N-1}) \cdot W),$$

$$\mathcal{H}_{:,i}^{var,N} = \mathcal{H}_{:,i}^{var,N}/||\mathcal{H}_{:,i}^{var,N}||_2,$$

$$(13)$$

$\mathcal{H}^{var}$ is the variable node feature; $\mathcal{HN}^{var}$ is the aggregation of neighbor variable node feature. $\mathcal{HN}^{var,N}_{:,i}$ aggregates both homogeneous and heterogeneous neighbor node feature of previous layer correlated to $v^{var}_i$. $\mathcal{H}^{var,N}_{:,i}$ is updated by the aggregated variable feature $\mathcal{HN}^{var,N}_{:,i}$ and its previous layer feature $\mathcal{H}^{var,N-1}_{:,i}$. Finally, the normalized $\mathcal{H}^{var,N}_{:,i}$ at $N$-th layer is the output of Cross-variable GNN.

## 3.4 Direct Multi-step forecasting

After obtaining the output features of Cross-Variable GNN, we exploit the direct multi-step (DMS) forecasting [33] for the decoder to predict the multi-step MTS at once. We take two MLPs as the decoder, where the first $\mathrm{MLP}_C$ maps the time dimension of features from $C$ to 1, while the second $\mathrm{MLP}_T$ maps the time dimension from the historical input sequence $L'$ to the output sequence length. The final prediction can be obtained by:

$$\{\hat{X}^t_1, ..., \hat{X}^t_D\}^{L+T}_{t=L+1} = \mathrm{MLP}_T(\mathrm{MLP}_C(\mathcal{H}^{var})). \tag{14}$$

Table 1: MTS forecasting results in terms of MSE and MAE, the lower the better. The prediction length $T \in \{96, 192, 336, 720\}$ and look back window size is set as 96. The best results are highlighted in **bold** and the second best are underlined.

| Models | | CrossGNN | | TimesNet | | Crossformer | | PatchTST | | ETSformer | | DLinear | | FEDformer | | Autoformer | | Pyraformer | | MTGNN | |
|---|---|---|---|---|---|---|---|---|---|---|---|---|---|---|---|---|---|---|---|---|---|
| Metric | | MSE | MAE | MSE | MAE | MSE | MAE | MSE | MAE | MSE | MAE | MSE | MAE | MSE | MAE | MSE | MAE | MSE | MAE | MSE | MAE |
| ETTm1 | 96 | **0.335** | 0.373 | 0.338 | 0.375 | 0.349 | 0.395 | 0.339 | 0.377 | 0.375 | 0.398 | 0.345 | **0.372** | 0.379 | 0.419 | 0.505 | 0.475 | 0.543 | 0.510 | 0.379 | 0.446 |
| | 192 | **0.372** | 0.390 | 0.374 | **0.387** | 0.405 | 0.411 | 0.376 | 0.392 | 0.408 | 0.410 | 0.380 | 0.389 | 0.426 | 0.441 | 0.553 | 0.496 | 0.557 | 0.537 | 0.470 | 0.428 |
| | 336 | **0.403** | **0.411** | 0.410 | 0.411 | 0.432 | 0.431 | 0.408 | 0.417 | 0.435 | 0.428 | 0.413 | 0.413 | 0.445 | 0.459 | 0.621 | 0.537 | 0.754 | 0.655 | 0.473 | 0.430 |
| | 720 | **0.461** | **0.442** | 0.478 | 0.450 | 0.487 | 0.463 | 0.499 | 0.461 | 0.499 | 0.462 | 0.474 | 0.453 | 0.543 | 0.490 | 0.671 | 0.561 | 0.908 | 0.724 | 0.553 | 0.479 |
| ETTm2 | 96 | **0.176** | **0.266** | 0.187 | 0.267 | 0.208 | 0.292 | 0.192 | 0.273 | 0.189 | 0.280 | 0.193 | 0.292 | 0.203 | 0.287 | 0.255 | 0.339 | 0.435 | 0.507 | 0.203 | 0.299 |
| | 192 | **0.240** | **0.307** | 0.249 | 0.309 | 0.263 | 0.332 | 0.252 | 0.314 | 0.253 | 0.319 | 0.284 | 0.362 | 0.269 | 0.328 | 0.281 | 0.340 | 0.730 | 0.673 | 0.265 | 0.328 |
| | 336 | **0.304** | **0.345** | 0.321 | 0.351 | 0.337 | 0.369 | 0.318 | 0.357 | 0.314 | 0.357 | 0.369 | 0.427 | 0.325 | 0.366 | 0.339 | 0.372 | 1.201 | 0.845 | 0.365 | 0.374 |
| | 720 | **0.406** | **0.400** | 0.408 | 0.403 | 0.429 | 0.430 | 0.413 | 0.416 | 0.414 | 0.413 | 0.554 | 0.522 | 0.421 | 0.415 | 0.433 | 0.432 | 3.625 | 1.451 | 0.461 | 0.459 |
| ETTh1 | 96 | **0.382** | **0.398** | 0.384 | 0.402 | 0.384 | 0.428 | 0.385 | 0.408 | 0.494 | 0.479 | 0.386 | 0.400 | 0.376 | 0.419 | 0.449 | 0.459 | 0.664 | 0.612 | 0.515 | 0.517 |
| | 192 | 0.427 | **0.425** | 0.436 | 0.429 | 0.438 | 0.452 | 0.431 | 0.432 | 0.538 | 0.504 | 0.437 | 0.432 | **0.420** | 0.448 | 0.500 | 0.482 | 0.790 | 0.681 | 0.553 | 0.522 |
| | 336 | 0.465 | **0.445** | 0.491 | 0.469 | 0.495 | 0.483 | 0.485 | 0.462 | 0.574 | 0.521 | 0.481 | 0.459 | **0.459** | 0.465 | 0.521 | 0.496 | 0.891 | 0.738 | 0.612 | 0.577 |
| | 720 | **0.472** | **0.468** | 0.521 | 0.500 | 0.522 | 0.501 | 0.497 | 0.483 | 0.562 | 0.535 | 0.519 | 0.516 | 0.506 | 0.507 | 0.514 | 0.512 | 0.963 | 0.782 | 0.609 | 0.597 |
| ETTh2 | 96 | **0.309** | **0.359** | 0.340 | 0.374 | 0.347 | 0.391 | 0.343 | 0.376 | 0.340 | 0.391 | 0.333 | 0.387 | 0.358 | 0.397 | 0.346 | 0.388 | 0.645 | 0.597 | 0.354 | 0.454 |
| | 192 | **0.390** | **0.406** | 0.402 | 0.414 | 0.419 | 0.427 | 0.405 | 0.417 | 0.430 | 0.439 | 0.477 | 0.476 | 0.429 | 0.439 | 0.456 | 0.452 | 0.788 | 0.683 | 0.457 | 0.464 |
| | 336 | **0.426** | **0.444** | 0.452 | 0.452 | 0.449 | 0.465 | 0.448 | 0.453 | 0.485 | 0.479 | 0.594 | 0.541 | 0.496 | 0.487 | 0.482 | 0.486 | 0.907 | 0.747 | 0.515 | 0.540 |
| | 720 | **0.445** | **0.464** | 0.462 | 0.468 | 0.479 | 0.505 | 0.464 | 0.483 | 0.500 | 0.497 | 0.831 | 0.657 | 0.463 | 0.474 | 0.515 | 0.511 | 0.963 | 0.783 | 0.532 | 0.576 |
| Electricity | 96 | 0.173 | 0.275 | 0.168 | 0.272 | 0.185 | 0.288 | **0.159** | **0.268** | 0.187 | 0.304 | 0.197 | 0.282 | 0.193 | 0.308 | 0.201 | 0.317 | 0.386 | 0.449 | 0.217 | 0.318 |
| | 192 | 0.195 | 0.288 | 0.184 | 0.289 | 0.201 | 0.295 | **0.177** | **0.278** | 0.199 | 0.315 | 0.196 | 0.285 | 0.201 | 0.315 | 0.222 | 0.334 | 0.378 | 0.443 | 0.238 | 0.352 |
| | 336 | 0.206 | 0.300 | 0.198 | 0.300 | 0.211 | 0.312 | **0.195** | **0.296** | 0.212 | 0.329 | 0.209 | 0.301 | 0.214 | 0.329 | 0.231 | 0.338 | 0.376 | 0.443 | 0.260 | 0.348 |
| | 720 | 0.231 | 0.335 | 0.220 | 0.320 | 0.223 | 0.335 | **0.215** | **0.317** | 0.233 | 0.345 | 0.245 | 0.333 | 0.246 | 0.355 | 0.254 | 0.361 | 0.376 | 0.445 | 0.290 | 0.369 |
| Traffic | 96 | **0.570** | **0.310** | 0.593 | 0.321 | 0.591 | 0.329 | 0.583 | 0.319 | 0.607 | 0.392 | 0.650 | 0.396 | 0.587 | 0.366 | 0.613 | 0.388 | 0.867 | 0.468 | 0.660 | 0.437 |
| | 192 | **0.577** | **0.321** | 0.617 | 0.336 | 0.607 | 0.345 | 0.591 | 0.331 | 0.621 | 0.399 | 0.598 | 0.370 | 0.604 | 0.373 | 0.616 | 0.382 | 0.869 | 0.467 | 0.649 | 0.438 |
| | 336 | **0.588** | **0.324** | 0.629 | 0.336 | 0.613 | 0.339 | 0.599 | 0.332 | 0.622 | 0.396 | 0.605 | 0.373 | 0.621 | 0.383 | 0.622 | 0.337 | 0.881 | 0.469 | 0.653 | 0.472 |
| | 720 | **0.597** | **0.337** | 0.640 | 0.350 | 0.620 | 0.348 | 0.601 | 0.341 | 0.632 | 0.396 | 0.645 | 0.394 | 0.626 | 0.382 | 0.660 | 0.408 | 0.896 | 0.473 | 0.639 | 0.437 |
| Weather | 96 | **0.159** | **0.218** | 0.172 | 0.220 | 0.191 | 0.251 | 0.171 | 0.230 | 0.197 | 0.281 | 0.196 | 0.255 | 0.217 | 0.296 | 0.266 | 0.336 | 0.622 | 0.556 | 0.230 | 0.329 |
| | 192 | **0.211** | 0.266 | 0.219 | **0.261** | 0.219 | 0.279 | 0.219 | 0.271 | 0.237 | 0.312 | 0.237 | 0.296 | 0.276 | 0.336 | 0.307 | 0.367 | 0.739 | 0.624 | 0.263 | 0.322 |
| | 336 | **0.267** | 0.310 | 0.280 | **0.306** | 0.287 | 0.332 | 0.277 | 0.321 | 0.298 | 0.353 | 0.283 | 0.335 | 0.339 | 0.380 | 0.359 | 0.395 | 1.004 | 0.753 | 0.354 | 0.396 |
| | 720 | 0.352 | 0.362 | 0.365 | 0.359 | 0.368 | 0.378 | 0.365 | 0.367 | 0.352 | 0.288 | **0.345** | 0.381 | 0.403 | 0.428 | 0.419 | 0.428 | 1.420 | 0.934 | 0.409 | 0.371 |
| Exchange | 96 | **0.084** | **0.203** | 0.107 | 0.234 | 0.097 | 0.214 | 0.108 | 0.223 | 0.085 | 0.204 | 0.088 | 0.218 | 0.148 | 0.278 | 0.197 | 0.323 | 1.748 | 1.105 | 0.102 | 0.228 |
| | 192 | **0.171** | **0.294** | 0.226 | 0.344 | 0.190 | 0.310 | 0.197 | 0.316 | 0.182 | 0.303 | 0.176 | 0.315 | 0.271 | 0.380 | 0.300 | 0.369 | 1.874 | 1.151 | 0.267 | 0.335 |
| | 336 | 0.319 | **0.407** | 0.367 | 0.448 | 0.362 | 0.429 | 0.375 | 0.429 | 0.348 | 0.428 | **0.313** | 0.427 | 0.460 | 0.500 | 0.509 | 0.524 | 1.943 | 1.172 | 0.393 | 0.457 |
| | 720 | **0.805** | **0.677** | 0.964 | 0.746 | 0.980 | 0.783 | 0.934 | 0.773 | 1.025 | 0.774 | 0.839 | 0.695 | 1.195 | 0.841 | 1.447 | 0.941 | 2.085 | 1.206 | 1.090 | 0.811 |

# 4 Experiment

## 4.1 Datasets And Experiment Setup

**Datasets** We conduct extensive experiments on 8 real-world datasets following [27], including **Weather**, **Traffic**, **Exchange Rate**, **Electricty** and 4 **ETT** datasets(ETTh1, ETTh2, ETTm1 and ETTm2). We follow the standard protocol in [27] and split datasets into training, validation and test set by the ratio of 6:2:2 for the last 4 ETT datasets, and 7:1:2 for the other datasets.

**Baselines and Setup** We compare our method with 9 state-of-the-art methods, including **Times-Net** [26]; 6 Transformer-based methods: **PatchTST** [12], **Crossformer** [34], **ETSformer** [25], **FEDformer** [35], **Pyraformer** [11], **Autoformer** [27]; GNN-based method: **MTGNN** [28]; simple yet powerful linear model **Dlinear** [33]. All the models are following the same experimental setup with prediction length $T \in \{96, 192, 336, 720\}$ for all datasets as in the original papers. We collect all baseline results from [26] except MTGNN, with the default look-back window $L = 96$. We

Table 2: Performance comparisons on ablative variants

| Datasets | | Weather | | | | Traffic | | | | ETTm2 | | | |
|---|---|---|---|---|---|---|---|---|---|---|---|---|---|
| Predict Length | | 96 | 192 | 336 | 720 | 96 | 192 | 336 | 720 | 96 | 192 | 336 | 720 |
| C-AMSI | MSE | 0.167 | 0.220 | 0.276 | 0.358 | 0.585 | 0.590 | 0.596 | 0.610 | 0.192 | 0.249 | 0.313 | 0.420 |
| | MAE | 0.229 | 0.284 | 0.318 | 0.382 | 0.330 | 0.336 | 0.341 | 0.354 | 0.279 | 0.318 | 0.358 | 0.417 |
| C-CS | MSE | 0.175 | 0.231 | 0.290 | 0.371 | 0.588 | 0.599 | 0.603 | 0.614 | 0.200 | 0.259 | 0.328 | 0.422 |
| | MAE | 0.241 | 0.286 | 0.325 | 0.385 | 0.331 | 0.343 | 0.342 | 0.358 | 0.281 | 0.325 | 0.364 | 0.421 |
| C-Hete | MSE | 0.172 | 0.224 | 0.277 | 0.364 | 0.589 | 0.588 | 0.603 | 0.616 | 0.190 | 0.255 | 0.320 | 0.417 |
| | MAE | 0.231 | 0.280 | 0.327 | 0.381 | 0.329 | 0.332 | 0.335 | 0.354 | 0.280 | 0.320 | 0.358 | 0.417 |
| C-CV | MSE | 0.174 | 0.227 | 0.279 | 0.367 | 0.588 | 0.591 | 0.604 | 0.616 | 0.193 | 0.257 | 0.322 | 0.419 |
| | MAE | 0.235 | 0.281 | 0.328 | 0.382 | 0.330 | 0.335 | 0.338 | 0.358 | 0.278 | 0.323 | 0.359 | 0.418 |
| CrossGNN | MSE | **0.159** | **0.211** | **0.267** | **0.352** | **0.570** | **0.577** | **0.588** | **0.597** | **0.176** | **0.240** | **0.304** | **0.406** |
| | MAE | **0.218** | **0.266** | **0.307** | **0.362** | **0.310** | **0.321** | **0.324** | **0.337** | **0.265** | **0.307** | **0.345** | **0.400** |

reproduced the results of MTGNN [28] with look-back window $L = 96$ on all datasets according to the settings in the original paper. We calculate the Mean Square Error(MSE) and Mean Absolute Error(MAE) of MTS forecasting as metrics. More details about datasets, baselines, implementation, hyper-parameters are shown in Appendix A.3.

## 4.2 Main Results

The quantitative results of MTS forecasting using different methods is shown in Table 1. CrossGNN achieves outstanding performance on most datasets across various prediction length settings, obtaining 47 first-place and 9 second-place rankings in total 64 settings. Quantitatively, compared with the best results that Transformer-based methods can offer, CrossGNN achieves an overall 10.43% reduction on MSE and 10.11% reduction on MAE. Compared with GNN-based method MTGNN, CrossGNN achieves a more significant reduction 22.57% on MSE and 25.74% on MAE. Compared with other strong baselines like TimesNet and Dlinear, CrossGNN can still outperform them in general. Our method dose not achieve the best performance on the Electricity dataset. Further analysis reveals that the more severe out-of-distribution (OOD) problem in Electricity dataset results in a lower generalization of the learned temporal graph relations on the test set.

## 4.3 Robustness Analysis of Noise

To evaluate the model robustness against noise, we add different intensities of Gaussian white noise to the original MTS and observe the performance changes of different methods. Figure 3 shows the MSE results of CrossGNN, ETSformer [25] and MTGNN [28] under different noise ratio on ETTm2 dataset, and the input length is set as 96. As the signal-to-noise ratio (SNR) decreases from 100db to 0db, the mean square error (MSE) increases more slowly on CrossGNN (0.177) compared to ETSformer (0.191) and MTGNN (0.205). The quantitative results demonstrate that CrossGNN exhibits good robustness against noisy data and has a great advantage when dealing with unexpected fluctuations. We speculate such improvements benefit from the explicit modeling of respective scale-level and variable-level interactions.

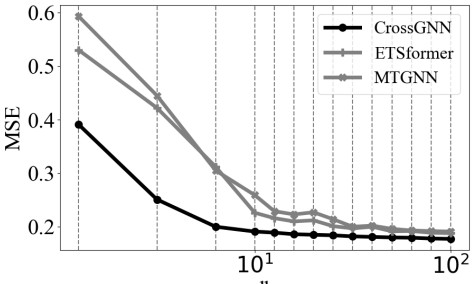

Figure 3: Robustness analysis under different signal-to-noise ratio (SNR) on ETTm2.

## 4.4 Ablation Study

We conduct ablation studies by removing corresponding modules from CrossGNN on three datasets. **C-AMSI** removes the adaptive multi-scale identifier (AMSI) and directly use $k$ fixed lengths (e.g., $1, 2, 3, ..., k$) for average pooling. **C-CS** removes the Cross-Scale GNN module. **C-Hete** removes the heterogeneous connections and focuses on the homogeneous correlation modeling between different variables. **C-CV** removes both homogeneous and heterogeneous connections. We analyze the results shown in Table 2. Obs.1) Removing

Cross-Scale GNN results in the most significant decrease in prediction metrics, emphasizing its strong ability in modeling the interaction between different scales and time points. Obs.2) Cross-Variable GNN also improves the model performance a lot, demonstrating the importance of modeling the complex and dynamic interaction between different variables. Obs.3) AMSI constantly improves the forecasting accuracy, suggesting that different scales of MTS contain rich interaction information.

### 4.5 Hyper-Parameter sensitivity

**Look-Back Window Size** Figure 4 shows the MSE results of models with different look-back window sizes on four datasets. As the window size increases, the performance of Transformer-based models fluctuates while CrossGNN constantly improves. This indicates that the attention mechanism of Transformer-based methods may focus much more on the temporal noise but our method can better extract the relationships between different time nodes through Cross-Scale GNN.

**Number of Scales** We vary the number of scales from 4 to 8 and report the MSE and MAE results on Weather and Traffic dataset. As shown in Figure 5 (a) and Figure 5 (b), We observe that the performance improvement becomes less significant after a certain number of scales, indicating that a certain scale size is sufficient to eliminate most of the effects of temporal noise. **Number of Node Neighbors** The number of neighboring nodes limited to each time node is mainly determined by the hyperparameter $K$. As shown on Figure 5 (c) and Figure 5 (d), we experiment with $K$ values of 10, 15, 20, 25, and 30 and found that CrossGNN is not sensitive to the number of $K$. This indicates that it is only necessary to focus on strongly correlated nodes for effective information aggregation in temporal interaction.

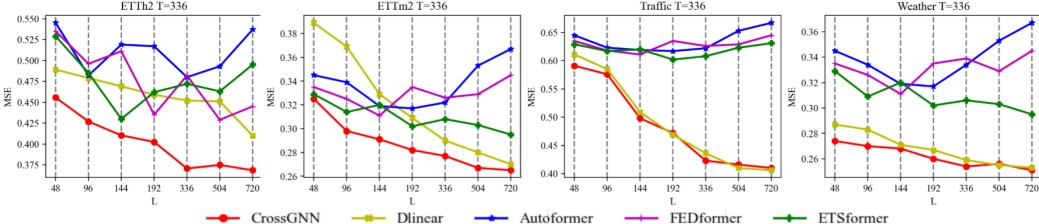

Figure 4: The MSE results (Y-axis) of models with different look-back window sizes (X-axis) on ETTh2, ETTm2, Traffic and Weather, the output length is set as 336.

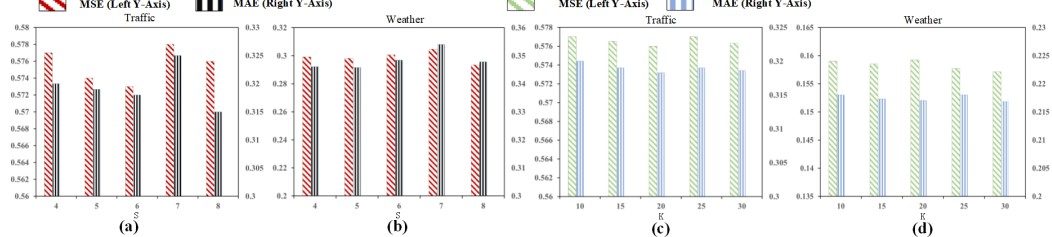

Figure 5: The MSE (left Y-axis) and MAE results (right Y-axis) of CrossGNN on Traffic and Weather. (a) and (b) display the performance on different scale numbers. (c) and (d) demonstrates the performance on different number of node neighbors

### 4.6 Complexity Analysis

Table 3 illustrates the theoretical complexity of CrossGNN and existing Transformer-based methods. Detailed complexity derivation can be found in the Appendix C. To verify that the time and space complexity of our method is indeed $O(L)$, we use TVM to implement the GNN computation part and compare the computation time and memory usage of with full connected graph during inference on ETTh2. Comparison experiments are implemented on a Intel(R) 8255C CPU @ 2.50GHZ with 40GB memory, centos 7.8, and TVM 1.0.0. Figure 6 illustrates the time and memory cost of GNN modules, and our proposed approach is close to linear with the input length.

| Method | Complexity per layer |
|---|---|
| Transformer | $O(L^2)$ |
| Informer | $O(L \log L)$ |
| Autoformer | $O(L \log L)$ |
| Pyraformer | $O(L)$ |
| CrossGNN | $O(L)$ |

Table 3: The computation complexity in theory. $L$ represents the length of the input data.

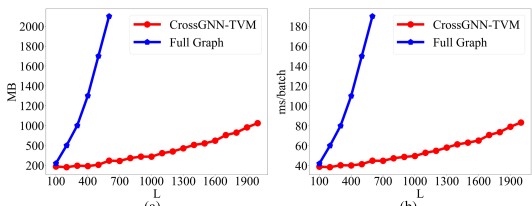

Figure 6: Comparison of time and memory consumption on ETTh2. (a) memory occupation. (b) computation time.

## 5 Conclusion and Future Work

In this work, we construct a comprehensive analysis of real-world data and observe that the temporal fluctuations and heterogeneity between variables, caused by unexpected noise, are not well handled by current popular time-series forecasting methods. To address above issues, we propose a linear complexity CrossGNN model, which is the first GNN model to refine both cross-scale and cross-variable interaction for MTS forecasting. An adaptive multi-scale identifier (AMSI) is leveraged to obtain multi-scale time series with reduced noise from input MTS. In particular, Cross-Scale GNN captures the scales with clearer trend and weaker noise, while Cross-Variable GNN maximally exploits the homogeneity and heterogeneity between different variables. Extensive experiments on 8 real-world MTS datasets demonstrate the effectiveness of CrossGNN over existing SOTA methods while maintaining linear memory occupation and computation time as the input size increases. For future work, it is worth exploring the design of dynamic graph networks that can effectively capture complex interactions in out-of-distribution (OOD) scenarios.

## 6 Acknowledgement

This paper is partially supported by the National Natural Science Foundation of China (No.62072427, No.12227901), the Project of Stable Support for Youth Team in Basic Research Field, CAS (No.YSBR-005), Academic Leaders Cultivation Program, USTC.

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

# Appendix

## A Experimental Details

### A.1 Datasets

We conduct extensive experiments on 8 real-world datasets following [27]. The interval length, time step number, and the variable number of each real-world dataset are presented in Table 4. The detailed dataset descriptions are as follows:

**1) ETTh (ETTh1, ETTh2, ETTm1, ETTm2)** consists of two hourly-level datasets (ETTh) and two 15minute-level datasets (ETTm). Each of them contains seven oil and load features of electricity transformers from July 2016 to July 2018.

**2) Weather** includes 21 indicators of weather, such as air temperature, and humidity. Its data is recorded every 10 min for 2020 in Germany.

**3) Traffic** describes hourly road occupancy rates measured by 862 sensors on San Francisco Bay area freeways from 2015 to 2016.

**4) Exchange-rate** collects the daily exchange rates of 8 countries from 1990 to 2016.

**5) Electricity** contains hourly electricity consumption (in Kwh) of 321 clients from 2012 to 2014.

Table 4: The statistics of the datasets for MTS forecasting.

| Datasets | ETTh1 | ETTh2 | ETTm1 | ETTm2 | Weather | Traffic | Exchange-rate | Electricity |
|---|---|---|---|---|---|---|---|---|
| Interval Length | 1 Hour | 1 Hour | 15 Minutes | 15 Minutes | 10 Minutes | 1 Hour | 1 Day | 1 Hour |
| Time step # | 17,420 | 17,420 | 69,680 | 69,680 | 52,696 | 17,544 | 7,588 | 26,304 |
| Variable # | 7 | 7 | 7 | 7 | 21 | 862 | 8 | 321 |

### A.2 Baseline Methods

We briefly describe the selected 9 state-of-the-art baselines as follows:

**1) TimesNet [26]** is a task-general foundational model for time series analysis that utilizes a modular architecture to unravel intricate temporal variations. A parameter-efficient inception block is leveraged to capture intra-period and inter-period variations in 2D space.

**2) Crossformer [34]** is a versatile vision transformer which solves attention among different variables.

**3) PatchTST [12]** is a strong baseline dependent on channel-independence and time series segmentation.

**4) ETSformer [25]** is a Transformer-based model for time-series forecasting that incorporates inductive biases of time-series structures and introduces novel exponential smoothing attention (ESA) and frequency attention (FA) to improve performance.

**5) DLinear [33]** is a simple linear-based model combined with a decomposition scheme.

**6) FEDformer [35]** is a Transformer-based model that uses the seasonal-trend decomposition with frequency-enhanced blocks to capture cross-time dependency for forecasting.

**7) Autoformer [27]** is a Transformer-based model using decomposition architecture with an auto-Correlation mechanism to capture cross-time dependency for forecasting.

**8) Pyraformer [11]** is a Transformer-based model learning multi-resolution representation of the time series by the pyramidal attention module to capture cross-time dependency for forecasting.

**9) MTGNN [28]** explicitly utilizes cross-variable dependency using GNN. A graph learning layer learns a graph structure where each node represents one variable in MTS.

### A.3 Implementation Details

To ensure a fair comparison, the look-back window size is set to 96, which is consistent with all baselines. We set the scale numbers $S$ to 5 and set $K$ to 10 for all datasets, as sensitivity experiments have shown that $S$ does not have a significant impact beyond 5 and CrossGNN is not sensitive to $K$. Additionally, the mean squared error (MSE) is used as the loss function. For the learning rate, a grid search is conducted among [5e-3, 1e-3, 5e-4, 1e-4, 5e-5, 1e-5] to obtain the most suitable learning rate for all datasets. Besides, the dimension of the channel is set as 8 for smaller datasets and 16 for larger datasets, respectively. The training would be terminated early if the validation loss does not

decrease for three consecutive rounds. The model is implemented in PyTorch 1.8.0 and trained on a single NVIDIA Tesla V100 PCIe GPU with 16GB memory.

# B  Additional Experimental Results

## B.1  Analysis on Robustness Against Noise

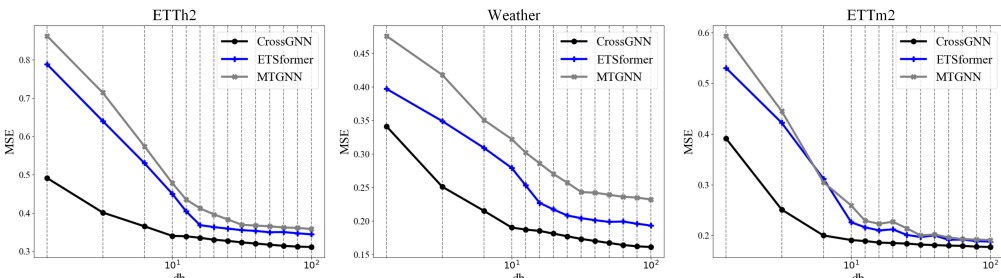

Figure 7: The MSE results (Y-axis) of models on ETTh2, ETTm2 and Weather with different signal-to-noise ratio (SNR).

To evaluate the robustness of CrossGNN against noise, we add different intensities of Gaussian white noise to the original MTS and observe the performance changes. As the intensity of Gaussian white noise increases, the signal-to-noise ratio (SNR) gradually decreases from 100 dB to 0 dB. Figure 7 shows the MSE results of CrossGNN, ETSformer [25] and MTGNN [28] on ETTh2, ETTTm2, and Weather under different SNR. As the SNR decreases from 100db to 0db, the mean square error (MSE) of CrossGNN increases more slowly than MTGNN and ETSformer.

Taking the results on the ETTm2 dataset as an example, when the noise intensity increases at the beginning (i.e., SNR decreases from 100db to 10db), the prediction accuracy of MTGNN and ETSformer becomes unstable. Their prediction accuracy drops more rapidly when the noise intensity suddenly increases (i.e., SNR decreases from 10db to 0db). In contrast, CrossGNN maintains overall stability, and its performance degrades more slowly. The quantitative results demonstrate that CrossGNN exhibits good robustness against noisy data and has a great advantage when dealing with unexpected fluctuations. Such improvements benefit from the explicit modeling of respective cross-scale and cross-variable interactions.

## B.2  Sensitivity Analysis

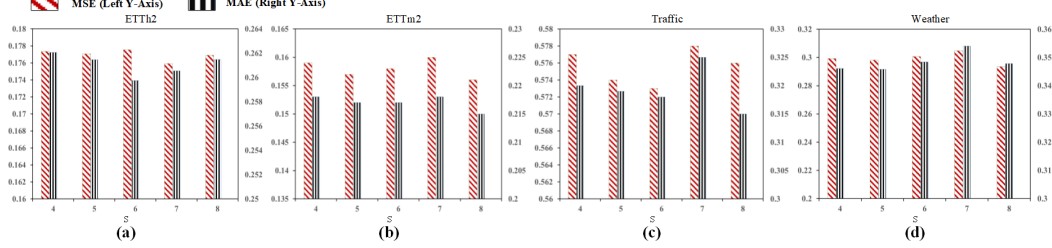

Figure 8: The MSE (left Y-axis) and MAE results (right Y-axis) of CrossGNN with different number of scales (X-axis) on ETTh2, ETTm2, Traffic, and Weather.

**Number of Scales**  We vary the number of scales from 4 to 8 and report the MSE and MAE results on ETTh2, ETTm2, Traffic, and Weather. As shown in Figure 8, We observe that the performance improvement becomes less significant after a certain number of scales (i.e., 5), indicating that a certain scale size is sufficient to eliminate most of the effects of temporal noise.

**Number of Temporal Node Neighbors**  The number of temporal neighboring nodes is primarily determined by the hyperparameter $K$. As depicted in Figure 9, we conducted experiments with

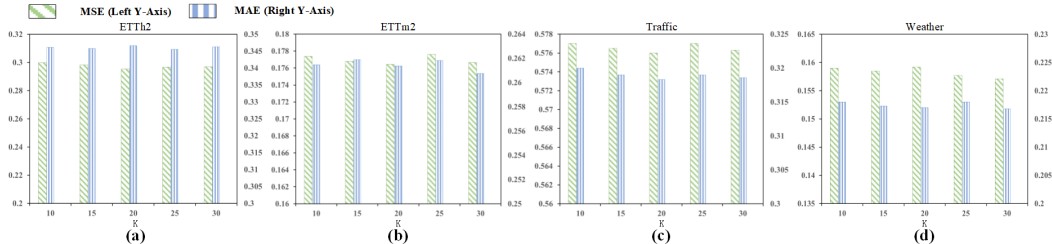

Figure 9: The MSE (left Y-axis) and MAE results (right Y-axis) of CrossGNN with different K (X-axis) on ETTh2, ETTm2, Traffic, and Weather.

different $K$ values, including 10, 15, 20, 25, and 30, and observed that CrossGNN is not sensitive to the number of $K$. This suggests that effective cross-scale interaction can be achieved by focusing only on strongly correlated time nodes.

### B.3 Visualization of Forecasting Results of Different Models

We present the visualization of forecasting results of CrossGNN and other baseline models on 8 datasets in Figure 10 and Figure 11. These datasets exhibit diverse temporal patterns, with 96-steps input length and output horizon. It can be observed that the prediction results of the Transformer-based model are significantly affected by noise, resulting in fluctuations. In contrast, the prediction results of CrossGNN are less affected by noise, and the predicted values are closer to the true results.

For example, considering the forecasting results on the Traffic dataset, there are three unexpected noise points (i.e., irregularly high points) in the input data. During prediction, the attention mechanism of the Transformer-based model may focus on the noisy points, leading to a bias towards higher output predictions. As a result, although the Transformer-based model seems to capture the periods of the time series, it fails to produce accurate predictions. In contrast, CrossGNN is unaffected by these three noisy data points and generates predictions that are closer to the ground truth. While Transformer-based models struggle to capture the scale and bias of future data due to unexpected noise in the input data, CrossGNN outperforms other models in terms of both scale and bias in forecasting.

## C   Derivation of Computational Complexity

In this section, we theoretically prove that the time and space complexity of the Cross-Scale module and Cross-Variable module in CrossGNN are both linear. We have organized the notations used in Table 5 for ease of reading.

Table 5: Meaning of notations

| Notation | Meaning |
|---|---|
| $v_i$ | The $i$-th time node |
| $S$ | Number of scales |
| $s$ | Index of the scale |
| $p_s$ | Period length of the $s$-th scale |
| $L$ | Original input length (i.e., look-back window size) |
| $L(s)$ | Time length at the $s$-th scale |
| $L'$ | Total length of concatenated multi-scale time series |
| $K^{scale}$ | The hyperparameter to control temporal neighbor numbers |
| $K^{var}$ | The hyperparameter to control variable neighbor numbers |
| $k_s$ | The temporal neighbor numbers at the $s$-th scale |
| $A(v_i)$ | Total temporal neighbor node number correlated to $v_i$ |
| $A$ | Total correlated temporal node pair number |
| $D$ | Variable numbers |

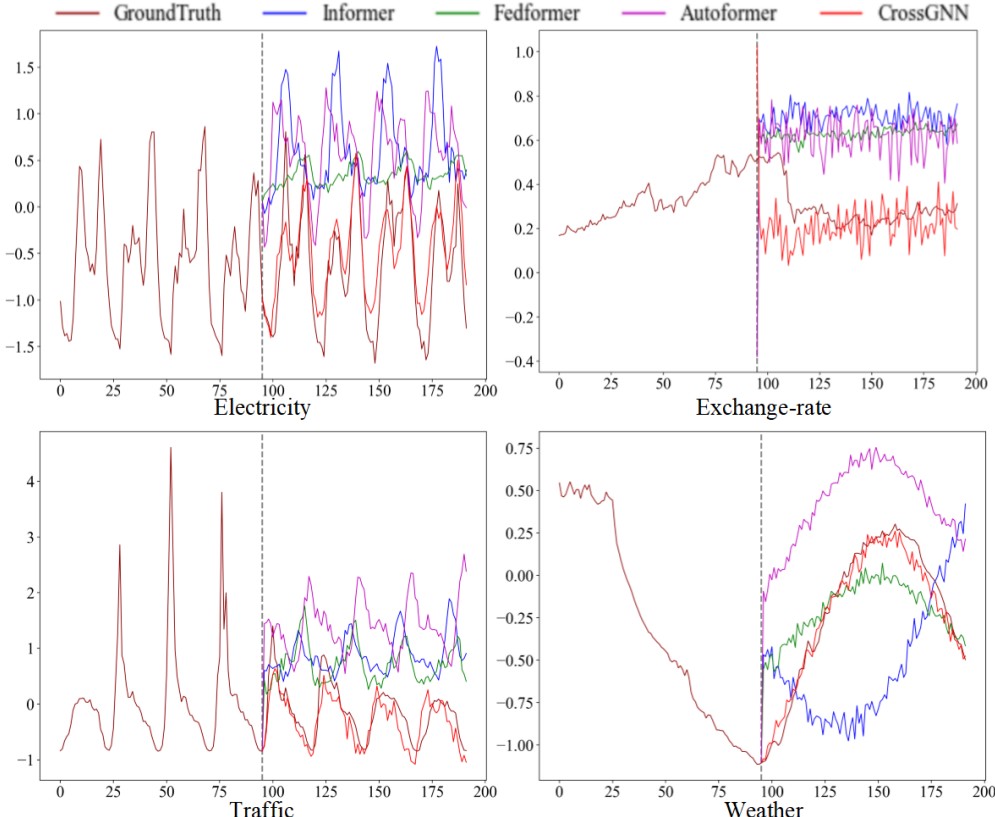

Figure 10: Visualization of 96-step forecasting results on Electricity, Exchange-rate, Traffic, and Weather, and the look-back window size is set as 96.

**Proposition 1.** *The time and space complexity for the Cross-scale GNN is $O(K^{scale} \times \ln S \times L)$ and amounts to $O(L)$ when $S$ and $K^{scale}$ are constants w.r.t. $L$.*

*Proof.* To improve readability, we substitute $K$ for $K^{scale}$. Denote $L(s)$ as the number of time nodes at $s$-th scale:

$$L(s) = \lfloor \frac{L}{p_s} \rfloor, 1 \le s \le S, \tag{15}$$

where $p_s$ is the corresponding period length of the $s$-th scale and $L$ is the original input length (i.e., look-back window size). $L'$ is the sum of time nodes at different scales, and it could be expressed by:

$$L' = \sum_{s=1}^{S} L(s) = \sum_{s=1}^{S} \lfloor \frac{L}{p_s} \rfloor \le \sum_{s=1}^{S} \lfloor \frac{L}{s} \rfloor \le L \sum_{s=1}^{S} \frac{1}{s} \approx L(\ln S + \epsilon + \frac{1}{2S}), \tag{16}$$

where $lnS + \epsilon + \frac{1}{2S} \approx \sum_{s=1}^{S} \frac{1}{S}$ is the approximate summation formula for the harmonic series, and $\epsilon$ is the Euler-Mascheroni constant.

For a time node, we set its scale-sensitive time node neighbor numbers to $k_s = \lceil \frac{K}{p_s} \rceil$ at $s$-th scale. Since the trend-aware neighbor nodes are defined as its previous node and next node at the current scale, the number of trend-aware neighbor nodes can reach 2 when these nodes do not overlap with the scale-sensitive neighbor nodes. However, when there is overlap, the number of trend-aware neighbor nodes can be 0 or 1. Therefore, the maximum neighbor node number of $v_i$ is given by:

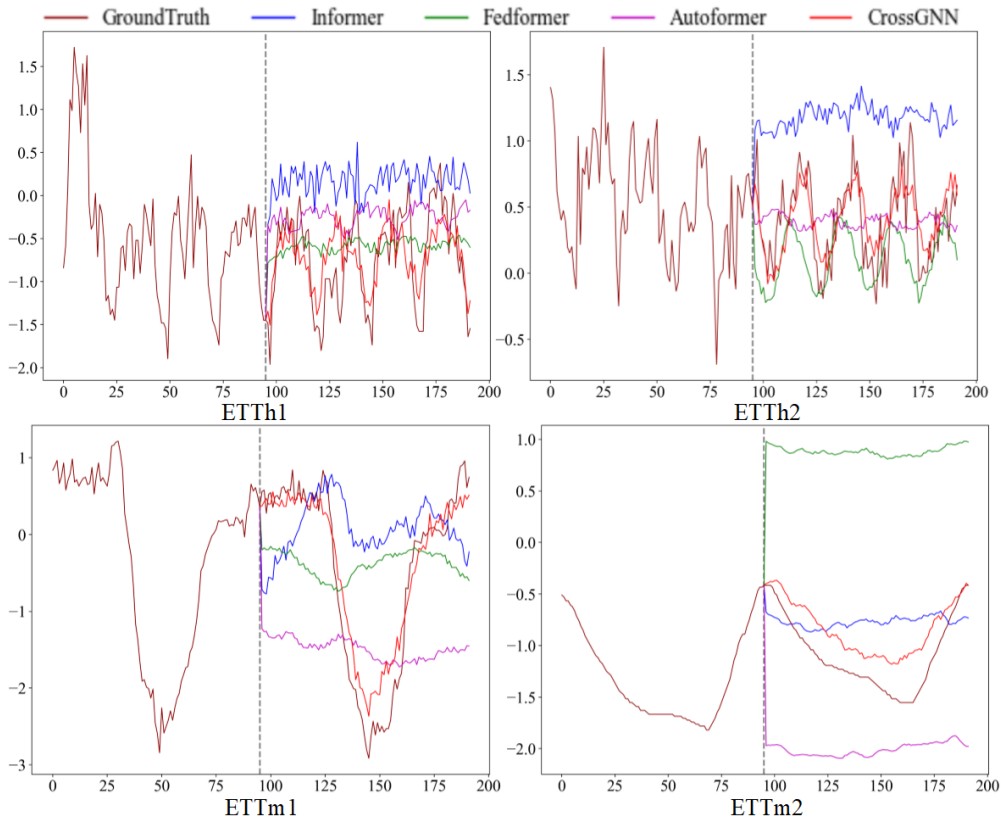

Figure 11: Visualization of 96-step forecasting results on ETTh1, ETTh2, ETTm1 and ETTm2, and the look-back window size is set as 96.

$$A(v_i) \le \sum_{s=1}^{S} k_s + 2 \tag{17}$$

$$= K \sum_{s=1}^{s=S} \frac{1}{p(s)} + 2 \tag{18}$$

$$\le K(\frac{1}{1} + \frac{1}{2} + ... + \frac{1}{S}) + 2 \tag{19}$$

$$\approx K(\ln S + \epsilon + \frac{1}{2S}) + 2, \tag{20}$$

Total correlated node pair number is expressed as:

$$A = L' \times A(v_i) \le L(\ln S + \epsilon + \frac{1}{2S})(K(\ln S + \epsilon + \frac{1}{2S}) + 2) \approx 2K \times \ln(S) \times L. \tag{21}$$

Consequently, the complexity of the proposed cross-scale GNN is:

$$O(A) \le O(2K \times \ln(S) \times L). \tag{22}$$

Since $K$ and $S$ are all constant terms that are independent of the length $L$ and remain fixed when $L$ changes, the complexity can be further reduced to $O(L)$. □

**Proposition 2.** *The time and space complexity for the Cross-variable GNN is $O(K^{var} \times D)$ and amounts to $O(D)$ when $K^{var}$ is a constant w.r.t. $D$.*

*Proof.* Without loss of generality, we assume that the number of homogeneous and heterogeneous correlated nodes for each variable are both $K^{var}$. For a cross-variable graph, there are a total of

$\sum_{i=1}^{D} 2K^{var} = 2K^{var}D$ correlated variable node pairs. Correspondingly, since the complexity of graph computation is related to the number of edges, the time and space complexity of cross-variable GNN are both $O(K^{var} \times D)$. As $K^{var}$ is a constant that is independent of $D$, its complexity is linear (i.e., $O(D)$). $\qquad\square$

