# OpenReview forum: "CrossGNN: Confronting Noisy Multivariate Time Series Via Cross Interaction Refinement"
_NeurIPS.cc/2023/Conference — NeurIPS 2023 poster_

### Official Review · Reviewer_LSa5 · 2023-06-24

**Soundness:** 3 good
**Presentation:** 3 good
**Contribution:** 3 good
**Rating:** 6
**Confidence:** 4

**Summary:**

This work first analyzes real-world datasets for multivariate time series forecasting and finds out two problems that are not well handled by previous works: 1) unexpected noise; 2) heterogeneity between variables. A GNN-based model, named CrossGNN, is proposed to fill the gap. CrossGNN consists of three components: 1)Adaptive Multi-Scale Identifier identifies potential periods with FFT and aggregates time series at different scales to construct a mult-scale view of the data; 2)Cross-Scale GNN constructs a graph among scales and uses GNN to capture cross-scale dependency; 3)Cross-Variable GNN captures homogeneous and heterogeneous dependency among variables. Correlation graphs used in Cross-Scale and Cross-Variable GNNs are restricted to be sparse, so the complexity is linear to the input length. Experiments on 8 real-world datasets demonstrate the effectiveness of the proposed model.

**Strengths:**

- This work is well-structured and easy to follow.
- The studied problem, noise in datasets, is an important and practical problem for MTS forecasting.
- Three components in CrossGNN are well-motivated and reasonable. The linear complexity is also an advantage.
- Robustness analysis of noise in section 4.3 is interesting.

**Weaknesses:**

- My main concern is that graphs used in Cross-Scale and Cross-Variable GNNs are static for each dataset. These graphs are constructed by shared learnable vectors with filtering, so they are static and the same for different inputs. This leads to 1) the dependency structure of time steps being fully determined by the positions in the series; 2) the dependency structure of variables being the same for different inputs in a dataset.
- Some recent works (e.g. PatchTST[1], Crossformer[2]) that explicitly model the cross-variable dependency should be compared in the main experiment.

[1]Yuqi Nie, Nam H. Nguyen, Phanwadee Sinthong, and Jayant Kalagnanam. A time series is worth 64 words: Long-term forecasting with transformers. In International Conference on Learning Representations, 2023.

[2]Yunhao Zhang and Junchi Yan. Crossformer: Transformer utilizing cross-dimension dependency for multivariate time series forecasting. In International Conference on Learning Representations, 2023.

**Questions:**

- In Equation (1), amplitudes are averaged over variables. Is the average operation reasonable? What if different variables have different periodic patterns?
- How is the ouput of AMSI (with shape $L' \times D$) embedded into latent vectors with shape $L' \times D \times C$?
- Settings of the ablation study should be described in detail, perhaps in the appendix. What does "divides the scales directly by fixed length" mean? Does C-Hete only remove the heterogeneous connections? If so, an ablation study that removes both homogeneous and heterogeneous connections should be conducted.
- It is better to compare the efficiency with baselines such as DLinear, Pyraformer, and TimesNet in Figure 6, as it is obvious that the proposed model is more efficient than the full graph. Moreover, the complexity of Pyraformer is $O(L)$, not $O(L \log L)$.
- See my other questions in the weakness part.

**Limitations:**

No potential negative societal impacts.

---

> ### Author Rebuttal · Authors · 2023-08-09
>
> Dear Reviewer LSa5,
>
> Thanks for your providing a positive feedback to our manuscript and encourage us to make further improvements. Now, we have addressed your concerns by supplementing both experimental studies and concise technical descriptions.
>
> **W1: Fixed Graph Structure for Different Inputs.**
> Based on your suggestions, we have analyzed the effects of a fixed dependency structure in CrossGNN from both temporal dimensions and variable dimension.
>
> - **Fixed Graph Structure in Temporal Dimension：**
> In a stable physical system, the relationship between time points is usually consistent, exhibiting a fixed change pattern over time. Such consistency can be influenced by noise, inducing the graph signal to be time-varying and dynamic with less regularity. Therefore, we are expected to capture the general and consistent relationship via GNN and learn a fixed temporal graph structure to avoid the impact of such noise.
>
> - **Fixed Graph Structure  in Variable Dimension：**
>   (1) A fixed dependency structure can effectively represent variable relationships across diverse inputs (remain robust under different input disturbance). And such effectiveness has been validated in MTGNN. (2) Our Cross-Variable GNN further decomposes variable-level relationships into positive and negative ones, enabling a more comprehensive and effective description of dependencies.   (3) Fixed graph structure can reduce forward computation time.
>
> We aim to learn a stable relationship between different time steps and variables within a fixed physical system (each dataset), reducing noisy effects on dependency structure. The concern you raised is the exact one we have considered in our work, yet the experimental results of CrossGNN can indicate that a fixed correlation matrix tends to be more effective compared to the variable correlation matrix. We appreciate your thoughtful analysis and raising this concern, we will place above analysis at the conclusion of our manuscript.
>
> **W2: Additional Comparative Experiments**
>
> Thank you for pointing out the experimental limitations. We worked day and night, conducting numerous experiments, and have documented the complete results in the **Table 1  and Table 2 of our submitted one-page PDF**.
>
> Experiments demonstrate that our method continues to exhibit excellent predictive capabilities compared to PatchTST and Crossformer. We appreciate your assistance and suggestions in enhancing the persuasiveness of our experiments, and we will include the complete results in the final version of our manuscript.
>
> **Q1: Averaged amplitudes.**
> There are two common strategies to achieve multi-scale time-series.
>
> 1. **As you mentioned**, each variable derives a variable-wise period based on its own frequency domain amplitudes. Then, each variable obtains distinct period length and subsequently forms variable-wise multi-scale MTS.
>
> 2. **In CrossGNN**, we average the frequency domain amplitudes of each variable and uniformly assign the same periods to all the variables, resulting in our multi-scale MTS.
>
> We take the latter strategy for following two reasons:
>
> 1. As most variables exhibit similar temporal patterns, the first strategy leads to less distinguishable amplitudes among variables.
>
> 2. Given comparable performance, the latter approach is more efficient in implementation, as it allows for parallel operations on all variables. We provide runtime and corresponding prediction metrics for the above two strategies in **Table 6 of the one-page PDF**  for your reference. The results demonstrate that averaging the amplitudes in CrossGNN achieves  comparable prediction performance while significantly reduces the running time.
>
>
>
> **Q2: How is $L^{\prime}\times D \rightarrow \ L^{\prime}\times D\times C$.**
>
>  $C$  refers to the (number of) channel dimension. Here, we employ an expansion dimension strategy (using an MLP), to create an embedding for each time step.  This strategy is inherited from MTGNN, aiming to enhance the local semantics at each time step and positively impact subsequent cross-scale and cross-variable interactions.  Thanks for you question, we will add the explanation in the final version.
>
>
>
> **Q3: Ablation Study.**
>
> Thank you for your valuable suggestion regarding ablations.
>
> 1. In AMSI, we learn $k$ potential periods of the sequences and perform average pooling based on these $k$ periods to obtain different scale representations. In **C-AMSI**, we remove the adaptive learning of period lengths and directly use $k$ fixed lengths (e.g., $1, 2, 3, ..., k$) for average pooling.
> 2. Thanks to the  insights of reviewers, we add a complete ablation study. **C-CV**: removing both homogeneous and heterogeneous connections. The results are provided in **Table 4 of the one-page PDF.** The results demonstrates that removing the entire Cross-Variable GNN module results in a significant performance degradation.
>
> We appreciate your suggestions for expression clarity and additional ablation experiments. We will incorporate these revisions  and additional results to our final version.
>
> **Q4: Training Efficiency & Complexity of Pyraformer.**
> 1. We have re-evaluated the time complexity of Pyraformer and confirmed it to be $O(L)$. Thanks for pointing it out and we will make the corrections in the corresponding section of the article.
> 2. Thank you for helping us enhance the validness of our paper. We comprehensively compare the per-batch running time of our model with well-acknowledged deep forecasting models (TimesNet, DLinear, Pyraformer). The results are conducted with the official model configuration, and are recorded in **Table 5 of the one-page PDF.** The results demonstrate CrossGNN the computation time of CrossGNN is significantly better than other models except for DLinear.
>
> Thank you again for your insightful suggestions and reviews.
>
> Authors of Paper 3421

---

> > ### Comment · Reviewer_LSa5 · 2023-08-11
> > **Response to Authors' Rebuttal**
> >
> > Thanks for your detailed response. I appreciate the thorough addressing of most of my concerns, including comparison with additional methods, averaged amplitudes, ablation study and efficiency analysis.
> >
> > The remaining concern is about graph structure learning: I agree that the graph structure among variables can be fixed as the relation is stable. But the fixed graph for temporal dimension appears somewhat unrealistic, especially since the graph is fully determined by the absolute positions in the input sequence. For example, assume the model tells us that timestamps 2 and 4 are connected in sequence 1. And we get sequence 2 by sliding the window one element to the right (sliding window is a common operation for training time series model). In this scenario, the fixed graph suggests a connection between 2 and 4, which corresponds to the relationship between 3 and 5 in the original sequence. This simple example shows that a static graph structure in the temporal dimension lacks practicality.
> >
> > Overall, the response makes this work more comprehensive and how to construct the graph for time series in temporal dimension is a complex and non-trivial open problem. I'll maintain my score of 6 for this work and am inclined to recommend that this paper be accepted.

---

> > > ### Author Response · Authors · 2023-08-11
> > >
> > > Dear reviewer LSa5,
> > >
> > > We would like to thank you for your encouraging follow-up.
> > > - Actually, the idea of a fixed temporal dependency can reasonably represent more stable relationship between temporal steps, and thus filter out the noise and unstable components, where the experimental results have also demonstrated such effectiveness. It is inspired by DLinear [1], which essentially learns a fixed temporal dependency structure by a simple linear layer.
> > > - Regarding the issue you raised that the input sequential order will impact the dependencies, it can be addressed by  a temporal positional embedding strategy, where we can train a temporal positional embedding and obtain the variable-wise adjacent matrix dependent on such trainable position embedding. When testing, we can look up to the embedding dictionary to dynamically construct the temporal point-wise correlations for a more flexible testing process. We will incorporate this positional embedding idea into our manuscript with both detailed strategy description and experimental results to further improve the scalability and flexibility of our solution.
> > >
> > > [1] Are transformers effective for time series forecasting? AAAI,2023.
> > >
> > > We sincerely thanks you for  your constructive advice and great help on our manuscript!
> > >
> > > Authors of Paper 3421

---

### Official Review · Reviewer_MKp9 · 2023-07-03

**Soundness:** 2 fair
**Presentation:** 2 fair
**Contribution:** 3 good
**Rating:** 5
**Confidence:** 3

**Summary:**

CrossGNN is a linear complexity GNN model designed for MTS forecasting, addressing two obstacles: self-attention mechanisms assigning high scores to outlier points and real-world data homogeneity and heterogeneity. By combining Adaptive Multi-Scale Identifier (AMSI), Cross-Scale GNN, and Cross-variable GNN, CrossGNN outperforms recent SOTA methods in real-world datasets.

**Strengths:**

1. Despite considering the relationship between variables, unlike Transformer-based models, it has a low time complexity O(L).
2. The paper also includes a comprehensive set of ablation studies and the performance in forecasting is SOTA.


**Weaknesses:**

1. The authors mentioned that noise can disturb attention mechanism in transformer. However, the noise ratio is less than 2\% as shown in Figure 1 (b) and I do not think this would be significant. Also, those outliers can be removed by using outlier detection techniques before training.
2. It is necessary to add an explanation or experiments to assure that the self-attention mechanism really adversely affects performance by assigning outlier points as high scores.
3. Technical novelty is a bit weak. Concepts of cross-scale and cross-variable are already introduced concepts in TimesNet and Crossformer. Also, many papers have already used GNN to deal with cross-variable interaction. What is the main difference except the input?

**Questions:**

In Figure 2, all scales are concatenated in time dimension from coarse to fine scale. Why this kind of concatenation is needed? Is there any reason for this?

**Limitations:**

There seems no limitation in this paper. Please include one.

---

> ### Author Rebuttal · Authors · 2023-08-09
>
> Dear Reviewer MKp9,
>
> Thank you for your insightful advice for polishing our manuscript. We have conducted sufficient experiments and analysis to dispel your concerns. The details can be found below.
>
> **W1: Noisy illustration issue in Figure 1 (b) & Whether it works when outliers removed.**
>
> (1) This is an illustrative example in Figure 1 (b). We allow the noise increase and present the varying results as illustrated in **Appendix B.1** to verify the effectiveness of CrossGNN. It demonstrates CrossGNN exhibits good robustness against noisy data and has a great advantage when dealing with unexpected noise.
>
>
>
> (2) The significance of denoise learning and reasons for not ruling out outliers directly：
>
> - **Noise can be inevitable in real world.** Figure 1 (b) provides an illustrative example based on four academic datasets. It is possible that the noise ratio in our actual application may be much higher than the reported 2%.
> - **Current anomaly detection algorithms may not necessarily detect all types of noise.** The noise can be summarized into two scenarios: persistence influence (e.g. the traffic accidents), instantaneous influence (e.g., sensors distortion). Some noise may exhibit sudden increases or decreases, while others may be continuous, low-amplitude variations.
> - **Directly filtering out outlier noise can result in significant information loss.** If we remove all the noisy samples detected by the noise detector, the remaining available samples may not provide sufficient support for training.
>
>
> In addition, we perform experiments by removing outlier points with noise detection algorithms [1] on four datasets. The prediction results are shown in **Table 3 of the one-page PDF**. We find that directly removing noisy samples have not improved the prediction performance.
>
> [1] Robust regression and outlier detection. 2009.
>
> **W2: Explanation and experiment of noise effect on self-attention.**
>
> Thank you for highlighting the importance of validating the impact of noise on the self-attention mechanism. We  have investigated the experimental results on four datasets and provide the explanation to your concerns from both  theoretical analysis and  empirical perspectives.
>
> **Theoretical analysis:**
> In the self-attention mechanism, for a given input time series data $X=[x_1,x_2,x_3,...,x_L]\in R^{L\times D}$, where $L$ denotes the sequence length and $D$ represents the embedding size, the attention score of the $i$-th time step with respect to the $j$-th time step can be expressed as $a_{i,j}=\frac{e^{(x_i\times q)\cdot (x_j \times k)}}{\sum_{k=1}^{L}e^{(x_i\times q)\cdot (x_k \times k)}}$, where $q\in R^{D \times C}$ and $k\in R^{D\times C}$ are the query and key matrices, respectively. It can be observed that the score $a_{i,j}$ depends on the value of $x_j$. When $x_j$ is a high-value outlier and the weights $q$ and $k$ are positive, $a_{i,j}$ will be assigned with a larger value.
>
> **Experimental results:**
>
> **(1) High score.** We trained predictive models using a Vallina Transformer on the ETTh1, Traffic datasets. **In Figure 1 of the one-page PDF,** the randomly sampled sequences and their corresponding attention matrices  demonstrate that applying the self-attention mechanism to time series data can indeed assign higher scores to outlier points.
>
> **(2) Performance  degradation.**  To evaluate the robustness of CrossGNN and transformer-based models against noise, we add different intensities of Gaussian white noise to the original MTS and observe the performance changes. **These results are in Appendix B.1.**  We found that by directly reducing the signal-to-noise ratio (increasing the noise intensity), the performance of the transformer model decreased rapidly, while CrossGNN decreased more slowly. It means noise will have a greater negative impact on the prediction performance of transformer-based models.
>
>
>
>
> We also observed in recent work [2] that visual Transformers might excessively focus on a few abnormal tokens, termed the "token overfocusing phenomena." This aligns with our perspective. We will include these experimental results in the final version, thanks for your suggestion to enhance the quality and validness of our manuscript.
>
> [2] Robustifying Token Attention for Vision Transformers. ICCV,2023.
>
>
>
> **W3: Technical novelty.**
>
> Thank you for your question. We have provided a detailed description of the technical novelties of CrossGNN in the global rebuttal and explained the differences between CrossGNN and other methods such as TimesNet, Crossform, and GNN-based.
>
> - **Time-interaction**.
>
> 1. **First attempt to explore GNN for temporal modeling**: Exploit GNN to connect different temporal points.
>
> 2. **First attempt to propose Cross-Scale information focus**: Refine interaction learning across different temporal scales to reduce noise effects.
>
> 3. **Adaptable Multi-Scale MTS Acquisition**: An  Adaptive Multi-Scale Identifier to dynamically partition scales based on sequence frequency characteristic.
>
> - **In variable interaction.**
>
> 1. **Pioneering Temporal Data Heterogeneity.**
>
> 2. **Low computation complexity.**
>
> These systematical novel solutions can jointly resolve the new problem of reducing noise in time-series and contribute to noisy-reduced learning scheme. Detailed descriptions of these technical novelty and differences from other models can be found in our common responses to all reviewers.
>
> **Q1: Concatenation of Scales.**
>
> The reasons can be three aspects.
>
> 1. The direct concatenation can well preserve the full multi-scale information.
>
> 2. To guarantee  cross-scale interaction, concatenating sequences of varying scales allows direct matrix multiplication for time-step level interactions between different scales.
>
> 3. This concatenation leverages the neuron connections and learnable weights of Fully Connected (FC) layers to facilitate feature extraction.
>
> Authors of Paper 3421

---

> > ### Comment · Reviewer_MKp9 · 2023-08-13
> > **Response to the Rebuttal**
> >
> > Thanks for your effort on the rebuttal. My concerns about noise are resolved, but I think technical novelty is still not enough.
> >
> > 1. Adaptive Multi-Scale Identifier exploits top-K frequency in Fourier Transform, which is very similar to temporal 2D-variation transformation of TimesNet. Plus, there is no justification why this kind of temporal scaling is selected. Is there any justification on selecting top-k frequency?
> >
> > 2. I doubt that this work is a combination of GNN and TimesNet. Is there any specialized component in GNN for this work?
> >
> > 3. The effect of GNN seems not in removing outliers as outlier ratio is small in evaluation datasets. If so, I do not understand the effect of modeling cross-scale interaction.

---

> > > ### Author Response · Authors · 2023-08-15
> > > **Response to Reviewer MKp9 1/2**
> > >
> > > Dear Reviewer MKp9,
> > >
> > > Thank you for reviewing our response and taking time to raise your questions. We will further elaborate on the novelty of our paper and provide detailed responses to your inquiries:
> > >
> > > **Q1. Justification on selecting top-k frequency.**
> > >
> > > **1.1 Top-k is necessary.**
> > >
> > > Actually, top frequency indicates the principal components in time-series, where we transfer the series expression from temporal  to frequency domain. Considering the sparsity of frequency domain and can avoid the noise brought by meaningless high frequencies \[1\]\[2\], we only select the top-$k$ amplitude values and obtain the most significant frequencies where $k$ is a hyperparameter. Through **hyperparameter experiments** on weather and traffic datasets (**in Figure 5(a) of the manuscript**), we found that setting $k$ as  5 achieves favorable results. Hence, we believe $k=5$ fits our model.
> > >
> > > **1.2 AMSI is novel and effective.**
> > >
> > > AMSI exploits top-$k$ frequency in Fourier Transform to capture periodicity. Our periodicity extraction module adaptively and efficiently captures different scales of time series by integrating an additional average pooling, which aggregates period information into scale-level. The combination of automatically obtained periods and average pooling mechanism is **a novel and effective approach to obtain multi-scale MTS**,  where it generates more suitable scales based on input data variations and better captures the evolving patterns of the input data.
> > >
> > > [1] The analysis of time series: an introduction. 1981.
> > >
> > > [2] FEDformer: Frequency enhanced decomposed transformer for long-term series forecasting. In ICML, 2022.
> > >
> > >
> > >
> > > **Q2. CrossGNN is not just combination of GNN and TimesNet**
> > >
> > > We will  clarify the distinctions from three aspects,  **1) Specialized component of CrossGNN**, **2) Differences from TimesNet**, **3) Differences from GNN-based models**.
> > >
> > > **2.1 Specialized component in GNN for this work**
> > >
> > > **We are the first to explore a lightweight and efficient pure GNN-based time series prediction model.**  Notably, we introduce novel designs in GNN modeling for time series data:
> > >
> > > 1. In the variable dimension, we are the first to separate variable relationships into homogeneous and heterogeneous graphs, enhancing the effectiveness of learning complex interactions.
> > > 2. In the temporal dimension, we are the first to apply GNNs for stable time relationship learning, focusing on associations across different scales.
> > > 3. In both time and variable dimensions, we employ pruning to remove less relevant edges, resulting in linear complexity for our GNN model.
> > >
> > >
> > >
> > > **2.2 Differences from GNN-based models.**
> > >
> > > GNN-based models typically operate on spatio-temporal data with pre-defined topological structures, leveraging pre-determined graph connections to learn spatial patterns and employing TCNs for temporal patterns. However, they overlook: **(1) heterogeneity among variables, (2) complexity of $O(N^2)$, without graph pruning for efficiency, (3) the potential of GNN-based approaches in capturing temporal relationships**, fail to exploit GNN to model series along temporal dimensions.
> > >
> > >
> > >
> > > **2.3 Differences from TimesNet.**
> > >
> > > CrossGNN is fundamentally distinct from TimesNet, with the only commonality being the utilization of frequency domain information to extract periods.
> > >
> > > - Although the solution to obtaining periods can  be replaced by techniques such as power spectral density diagrams and autocorrelation analysis, we choose FFT for obtaining periods as it is faster and its effectiveness has been validated in other works [3].
> > >
> > > - Regarding differences, in AMSI, we extract periods adaptively to capture diverse time scales through average pooling, fostering interaction between different scales. In addition, we capture the heterogeneous interactions among variables via extracting stable relations through GNN, and achieve a lightweight model through pruning. However, in contrast, TimesNet transforms sequences into 2-D inputs for large CNN-based backbones, limited to capturing adjacent period information and incurring substantial memory and computational costs.
> > >
> > > Furthermore, we provide  comparisons on two datasets of memory usage, runtime, and MSE metric as follows:
> > >
> > > **ETTm2:**
> > >
> > > | Model            | CrossGNN     | TimesNet |
> > > | ---------------- | ------------ | -------- |
> > > | Time (per batch) | **15.6 ms**  | 205.4 ms |
> > > | Memery           | **1.367 GB** | 3.285 GB |
> > > | MSE              | **0.309**    | 0.340    |
> > >
> > > **Weather:**
> > >
> > > | Model            | CrossGNN     | TimesNet |
> > > | ---------------- | ------------ | -------- |
> > > | Time (per batch) | **20.4 ms**  | 309.3 ms |
> > > | Memery           | **2.674 GB** | 5.727 GB |
> > > | MSE              | **0.159**    | 0.172    |
> > >
> > > These results show that CrossGNN reduces  running time by more than 10 times and saves memory by more than half, while the prediction performance is better.
> > >
> > > [3] Autoformer: Decomposition transformers with auto-correlation for long-term series forecasting. NeurIPS, 2021.

---

> > > > ### Author Response · Authors · 2023-08-15
> > > > **Response to Reviewer MKp9 2/2**
> > > >
> > > > **3 Modeling cross-scale interaction is necessary**
> > > >
> > > > The effectiveness of modeling cross-scale interactions is demonstrated in two aspects:
> > > >
> > > > 1) **Regarding noise**, cross-scale interactions allow the model to obtain signals with lower noise intensity from coarser temporal scales. We experimentally show that the Cross-Scale module effective in mitigating outliers. **In Appendix B.1, we increase noise intensity on ETTh2, ETTm2, and Weather datasets.** The results indicate that compared to other  methods (Transformer-Based and GNN-Based), CrossGNN exhibits more effective and stable performances under enhanced noise conditions. **This verifies the efficacy of Cross-Scale GNN in combating outliers.**
> > > >
> > > > 2) **Addressing the inherent correlations across different scales.** Interactions and dependencies exist among various scales. Coarse granularity captures the temporal trends, while fine granularity (local patterns) refines the expression of coarse-grained information. Therefore, cooperative learning through modeling cross-scale interactions enhances mutual robustness.
> > > >
> > > > Furthermore, **in the ablation experiment  (Table 2 in the manuscript), removing cross-scale interactions leads to an average performance drop of 5.2% for CrossGNN** across the three datasets. **This further verifies the efficacy of modeling cross-scale interactions in enhancing predictive performance.**
> > > >
> > > > Authors of Paper 3421

---

> > > > > ### Comment · Reviewer_MKp9 · 2023-08-15
> > > > > **Response to Authors**
> > > > >
> > > > > Thanks for pointing out the difference between the paper and existing works related to GNN and TimesNet. CrossGNN has its own empirical novelty and is effective in MTS forecasting. However, it still lacks theoretical background in AMSI, many part of techniques seems not novel, and performance margin is not that big. In conclusion, I raised my score to 5.

---

> > > > > > ### Author Response · Authors · 2023-08-17
> > > > > > **Thanks for your positive feedback**
> > > > > >
> > > > > > Dear Reviewer MKp9,
> > > > > >
> > > > > > We gratefully appreciate the   positive follow-up, and we will take further efforts in improving the clarification of our proposed AMSI in the next revision and provide more theoretical justification of frequency-perspective series learning, such as adding the foundation of spectrum theory and Fast Fourier Transformation (FFT) in series perspective.  Thanks again!
> > > > > >
> > > > > >
> > > > > > Authors of Paper 3421

---

### Official Review · Reviewer_zRqD · 2023-07-04

**Soundness:** 2 fair
**Presentation:** 3 good
**Contribution:** 3 good
**Rating:** 6
**Confidence:** 4

**Summary:**

Overall Comment：
This article addresses two issues in multi-variate time-series modeling: i) How to address signal noise in multivariate time series, and ii) How to address interactions between multiple variables to extract information. The article proposes two GNN models to solve these problems, including the Cross-Scale GNN model for addressing signal noise and the Cross-Variable GNN for addressing interactions between different signals.

Overall, while some parts of the article are not very clear, such as, what is the meaning of  homogeneity and heterogeneity in time series, and the the ablation study can be improved, the article's innovative ideas are clear and worth being seen by more people. Compared to conventional baseline methods, this article has significant advantages.

The strengths of the article include:

1. The paper conducts a large number of experiments, tested on 8 datasets. From the experimental results, the proposed method is effective and performs better than conventional MTS baseline methods.
2. The article is well written and easy to read, making it easy to understand the information that the author is trying to convey.

The weaknesses of the article include:

1. The analysis of the ablation study needs to clarify that which module, the Cross-Scale GNN and Cross-Variable GNN, plays a larger role in the model in order to judge whether it is filtering signal noise that makes the proposed model perform better on each dataset or whether introducing interactions between multiple variables is the key factor.
2. Correspondingly, if the Cross-Scale GNN model for addressing signal noise plays a larger role, is the focus of the article on time series denoising or modeling time series with noise? The article should be compared with methods specifically designed for denoising time series, rather than just conventional time series modeling baselines.
3. What is the meaning of homogeneity and heterogeneity in time series signals? In Figure 1(d), only positive or negative correlations between variables are displayed, which is not related to homogeneity and heterogeneity relationships between variables.

**Strengths:**

The strengths of the article include:

1. The paper conducts a large number of experiments, tested on 8 datasets. From the experimental results, the proposed method is effective and performs better than conventional MTS baseline methods.
2. The article is well written and easy to read, making it easy to understand the information that the author is trying to convey.

**Weaknesses:**

The weaknesses of the article include:

1. The analysis of the ablation study needs to clarify that which module, the Cross-Scale GNN and Cross-Variable GNN, plays a larger role in the model in order to judge whether it is filtering signal noise that makes the proposed model perform better on each dataset or whether introducing interactions between multiple variables is the key factor.
2. Correspondingly, if the Cross-Scale GNN model for addressing signal noise plays a larger role, is the focus of the article on time series denoising or modeling time series with noise? The article should be compared with methods specifically designed for denoising time series, rather than just conventional time series modeling baselines.
3. What is the meaning of homogeneity and heterogeneity in time series signals? In Figure 1(d), only positive or negative correlations between variables are displayed, which is not related to homogeneity and heterogeneity relationships between variables.

**Questions:**

As I write in the Weaknesses:

1. the Cross-Scale GNN and Cross-Variable GNN, which plays a larger role in the model？ Whether it is filtering signal noise that makes the proposed model perform better on each dataset or whether introducing interactions between multiple variables is the key factor?

2. What is the meaning of homogeneity and heterogeneity in time series signals? In Figure 1(d), only positive or negative correlations between variables are displayed, which is not related to homogeneity and heterogeneity relationships between variables.

**Limitations:**

The authors didn't address the limitations clearly.

---

> ### Author Rebuttal · Authors · 2023-08-09
>
> Dear reviewer zRqD,
>
> Thank you for your valuable insights  for polishing our manuscript. We have conducted additional experiments and analysis to address your concerns.
>
> **W1&Q1 Which module is more important.**
>
> Thank you for your question in our experimental analysis. Our ablation experiments in manuscript section 4.4 have revealed that the Cross-Scale module holds greater significance. Across twelve settings in three datasets, the removal of Cross-Scale GNN resulted in performance degradation in all settings, positioning it at the bottom of the rankings in ten settings. This pivotal finding emphasizes the effect of Cross-Scale GNN on modeling temporal relationships, effectively filtering signal noise, and enhancing downstream prediction outcomes. In the final version, we will emphasize the importance of the Cross-Scale module in the Ablation section, acknowledging your guidance in polishing our experimental analysis.
>
> **W2: Add comparative baselines & Robust analysis on noise.**
>
> Thank you for your valuable suggestions.
>
> - We have incorporated comparisons with denoising time series prediction methods, such as Stationary (a model designed for unstable time series), and presented the experimental results **in Table 1 and Table 2 of one-page PDF.** The  results demonstrate that our CrossGNN still outperforms other denosing time series prediction methods.
>
> - Additionally, to validate CrossGNN performance in time series prediction with noisy data, we introduced varying levels of Gaussian white noise to three datasets (ETT2, ETTm2, and Weather). The experimental outcomes in **the Appendix B.1** illustrate that  as the intensity of Gaussian white  noise increases, the signal-to-noise ratio (SNR) gradually decreases from 100 dB to 0 dB. As the SNR decreases from 100db to 0db, the mean square error (MSE) of CrossGNN increases more slowly than MTGNN and ETSformer. The quantitative results demonstrate that CrossGNN exhibits good robustness against noisy data and has a great advantage when dealing with unexpected fluctuations. We speculate such improvements benefit from the explicit modeling of respective scale level and variable-level interactions.
>
> **W3&Q2: Meaning of homogeneity and heterogeneity.**
>
> Thank you for your advice in improving our expression. In temporal signals, heterogeneity indicates that the temporal patterns between variables are not similar, while homogeneity indicates similar temporal patterns between variables. In the manuscript,  we use negative correlation to represent heterogeneous variables and positive correlation to represent homogeneous variables.
>
> Thanks for providing the encouraging reviews and your valuable suggestions indeed make sense to further improve our manuscript. Thanks again!
>
> Authors of Paper 3421

---

> > ### Comment · Area_Chair_eVcZ · 2023-08-15
> > **Requesting an update from Reviewer zRqD following the authors' rebuttal**
> >
> > Reviewer zRqD,
> >
> > As the discussion period is nearing its end, please read the authors' response to your comments on Paper 3421 and indicate whether your concerns are addressed.

---

> > > ### Comment · Reviewer_zRqD · 2023-08-20
> > > **Reply to author rebuttal**
> > >
> > > My concerns about the work are mostly solved. I tend to accept this work.

---

> > > > ### Author Response · Authors · 2023-08-20
> > > > **Thanks for positive follow-up**
> > > >
> > > > Dear Reviewer zRqD,
> > > >
> > > > We sincerely appreciate your constructive feedback and valuable comments, which really contributed to the enhancement of our manuscript.  Thanks very much!
> > > >
> > > > Authors of Paper 3421

---

> > > > > ### Author Response · Authors · 2023-08-21
> > > > > **Thanks again!**
> > > > >
> > > > > Dear Reviewer zRqD,
> > > > >
> > > > > As the author-reviewer discussion stage is ending soon, we would like to kindly inquire whether our addressing of concerns might potentially lead to re-evaluating our paper. Thanks again for your valuable time and insightful comments on our manuscript!
> > > > >
> > > > > Authors of Paper 3421

---

### Official Review · Reviewer_F3ix · 2023-07-06

**Soundness:** 2 fair
**Presentation:** 3 good
**Contribution:** 2 fair
**Rating:** 5
**Confidence:** 5

**Summary:**

This paper aims to deal with the temporal fluctuations and heterogeneity between variables, caused by unexpected noise, for better multivariate time-series forecasting. Specifically, the authors propose a linear complexity CrossGNN model, including Cross-Scale GNN which captures relationships inter- and intra- scales, and Cross-Variable GNN which captures the homogeneity and heterogeneity relationships between different variables. Experiments on 8 benchmark multivariate time-series datasets demonstrate the effectiveness of CrossGNN over some existing methods.

**Strengths:**

1. The authors propose a GNN-based (CrossGNN) method with linear complexity for long-term time series forecasting.
2. The CrossGNN captures the relationships between both scales and variables.
3. The paper is easy to understand.

**Weaknesses:**

1. The proposed CrossGNN does not compare with the SoTA methods, like RLinear, RMLP [1] and PatchTST [2]. I checked the results in this paper (Table 1 & Figure 4) and [1,2], and found that CrossGNN is worse than RLinear, RMLP and PatchTST.

[1] Revisiting Long-term Time Series Forecasting: An Investigation on Linear Mapping

[2] A Time Series is Worth 64 Words: Long-term Forecasting with Transformers

2. It is unclear why the heterogeneity between variables is caused by unexpected noise.
3. In the top of Page 5, it is unclear why the production of two learnable vectors can diminish the effect of noise.
4. What is the meaning of "C" in Figure 2?

**Questions:**

1. Please compare with SoTA methods, like RLinear, RMLP and PatchTST.
2. Please give explanations on the weaknesses 2~4.

**Limitations:**

Yes

---

> ### Author Rebuttal · Authors · 2023-08-09
>
> Dear Reviewer F3ix,
>
> Thanks for your valuable comments for improving our manuscript. Firstly, we have incorporated a comparison with the SoTA baselines. Subsequently, we have seriously addressed and clarified the concerns  you raised as below.
>
> **W1: Lack of SoTA baselines.**
> Based on your suggestion, we are working diligently to conduct experiments and have added the abovementioned three SoTA baselines for comparision  **in Table 1 and Table 2 of the one-page PDF**.
>
> Nevertheless, directly comparing  our solution with  RMLP, RLinear, PatchTST may be unfair.
> The reasons can be two aspects,
>
> 1. **Publication date.** The publication date of RLinear, RMLP (18th, May) on arxiv is after the  deadline (17th, May) of NeurIPS.
>
> 2. **Different experimental settings.** Our CrossGNN takes 96 time steps as input while the input of RMLP, RLinear, PatchTST are 336, which are longer than ours. It is obvious that much longer sequence can lead to higher performance but deterioriates the efficiency.
>
>
> Finally, we emphasize that we now conduct fair comparison experiments in the following ways:
>
>  - **Input length 96:**  We directly compared CrossGNN with PatchTST, RMLP, RLinear with the same input length.
>  - **Input length 336:** Similar to RMLP, RLinear, and PatchTST, we also incorporated the **RevIN** [3] technique in CrossGNN to mitigate data shift effects.
>
> The final experimental results (**in Table 1 and Table 2 of the one-page PDF**) demonstrate that our CrossGNN still outperforms recent SoTA methods, simultaneously maintaining linear complexity.
>
> Thank you again for suggestions in improving our experiments. We will include the complete experimental results in the final version and cite the two referenced papers  \[1\]\[2\] accurately.
>
> [1] Revisiting Long-term Time Series Forecasting: An Investigation on Linear Mapping
>
> [2] A Time Series is Worth 64 Words: Long-term Forecasting with Transformers
>
> [3] Reversible instance normalization for accurate time-series forecasting against distribution shift. ICLR, 2022.
>
>
>
>
> **W2: Heterogeneity between variables caused by unexpected noise.**
> **Heterogeneity between variables:**  Referring to some variables exhibiting distinct temporal patterns.  In a stable physical system, most variables exhibit homogeneous correlations. Heterogeneity generally arises from unexpected noise, this can be summarized into two scenarios:
>
> **(1) Persistent influence:** Continuous noise, such as emergency events, leads to sustained changes in the temporal patterns of certain variables, resulting in disparate time series. For instance, in the Traffic dataset, some traffic intersections exhibit higher nighttime traffic than daytime due to accidents during the day.
>
>  **(2) Instantaneous influence:** Brief yet high-frequency noise, like sensor distortions, introduces irregular data points into the time series of certain variables, reducing the overall sequence regularity.
>
> The above kinds of noise can affect the time signal as two aspects, 1) different variables must exhibit various evolutionary patterns, 2)  the correlations between variables are varying along time，thus we design Cross-Variable module to effectively enhance the role of cross-variable relationship learning in time series prediction.
>
>
>
>
>
>
>
>
>
> **W3: The production of two learnable vectors.**
>
> Eq(6) at the top of Page 5 aims at learning stable and cross-scale temporal correlations $E^{scale}\in R^{L'\times L'}$ not affected by input noise. Here, we explain why it can mitigate the impact of input noise on learning temporal correlations:
>
> - **Input-independent**: Previous works learns the correlation weight by considering the correlation as the function of input $X$ while in this paper, we initialize the learnable correlation $E^{sacle}$  as the production of two learnable vectors $vec_1^{scale}$ and $vec_2^{scale}$, which are independent of $X$. Thus, it cannot be disturbed by noisy $X$ and directly learn the correlation regularity from general data sequences.
> - **Cross-scale**: We extend $E^{scale}$  into multiple scales with coarse sequence, as shown in main text Figure 1(c), where the insight is that coarse temporal scale extracts the backbone of regularity and thus less influenced by noise.
>
> Besides, we can also verify the significant motivation of our denoising insight from the analysis of self-attention. Through extensive experiments of noise impact on learning long temporal correlations, we observed cases where abnormal noise received excessive attention, as depicted in **Figure 1 of one-page PDF**. Thus, exploring dependency structure independent of inputs noise  is crucial.
>
>
>
> **W4: The meaning of "$C$" in Figure 2.** $C$  refers to the (number of) channel dimension. Here, we employ an expansion dimension strategy, creating an embedding for each time step.  This strategy is inherited from MTGNN [4], aiming to enhance the local semantics at each time step and positively impact subsequent cross-scale and cross-variable interactions.  Thanks for you question, we will add the explanation in the final version.
>
> [4] Multivariate time series forecasting with graph neural networks. SIGKDD, 2020.
>
> Authors of Paper 3421

---

> > ### Comment · Reviewer_F3ix · 2023-08-15
> >
> > Thank you very much for your responses.
> > Is there any reason why only the results of 4 datasets are shown in Table 1 and Table 2 of the one-page PDF?
> > For Input length 336, is it possible to show the results of CrossGNN without RevIN? It is better to understand the performance improvement comes from the proposed CrossGNN or RevIN, when comparing with PatchTST.

---

> > > ### Author Response · Authors · 2023-08-17
> > > **Response to  follow-up questions of Reviewer F3ix (Part 1/3)**
> > >
> > > Dear Reviewer F3ix,
> > >
> > > Thank you for taking the time with further discussions to polish our  manuscript. At this time, we provide more detailed clarification and experiments to address concerns you've raised.
> > >
> > > **1. Only 4 datasets  are recorded in the one-page PDF.**
> > >
> > > Due to space limitations, our initial response only provided empirical results on four representative datasets: two smaller ETT datasets, ETTh2 and ETTm2, and two larger datasets, Weather and Traffic. The results demonstrate the performance of CrossGNN still outperforms RLinear, RMLP, and PatchTST under equivalent settings. In this report, we present a more comprehensive evaluation of CrossGNN against PatchTST across all eight datasets.
> > >
> > > **2. Analysis on the comparison between PatchTST and CrossGNN**
> > >
> > > We set the input length to 336 and compare the performance of PatchTST (w,w/o RevIN) and CrossGNN (w, w/o RevIN) on following eight datasets. The following observations are made:
> > >
> > > **2.1 Forecasting performance：**
> > >
> > > **(1) w/o RevIN on two models, CrossGNN demonstrates an average performance   8.36%  gain on MSE and 5.8% gain on MAE against PatchTST. (2) When both models incorporating RevIN, CrossGNN exhibits an average performance improvement of 3.42% MSE and 2.5% MAE against PatchTST.** This highlights not only the superior predictive capabilities of CrossGNN but also its advantage in handling sequence anomalies (considering the instance normalization nature of RevIN). Moreover, as the reported results of PatchTST are based on RevIN, we directly compare the performance of CrossGNN without RevIN against PatchTST with RevIN on the settings of 64 settings on 8 datasets. **The results show that in 50/64 settings, CrossGNN without RevIN still outperforms PatchTST with RevIN.** This verifies the inherent superiority of CrossGNN in predictive performance  compared to PatchTST.
> > >
> > > **2.2 Computational Efficiency:**
> > >
> > > We record the per-batch runtime and memory usage for PatchTST and CrossGNN under identical settings. **The results reveal that  CrossGNN not only outperforms PatchTST in performance, but also achieves 10 times faster runtime and 8 times less memory consumption than PatchTST.** This advantage is attributed to the lightweight pruning employed in GNN..
> > >
> > >
> > >
> > > **The detailed results are as below.**
> > >
> > >
> > > **Table 1. Efficiency on 8 datasets.**
> > >
> > > | PatchTST    | ETTh1 | ETTh2 | ETTm1 | ETTm2 | Traffic |  Elec  | Weather | Exchange |
> > > | ----------- | :---: | :---: | :---: | :---: | :-----: | :----: | :-----: | :------: |
> > > | Time (ms)   | 54.0  | 53.9  | 54.1  | 53.8  | 1692.2  | 191.7  |  141.0  |   75.7   |
> > > | Memory (GB) | 3.662 | 3.645 | 3.647 | 3.634 | 79.674  | 39.162 |  4.732  |  3.837   |
> > >
> > > | CrossGNN    | ETTh1 | ETTh2 | ETTm1 | ETTm2 | Traffic | Elec  | Weather | Exchange |
> > > | ----------- | :---: | :---: | :---: | :---: | :-----: | :---: | :-----: | :------: |
> > > | Time (ms)   | 15.6  | 15.7  | 15.4  | 15.7  |  84.3   | 59.5  |  21.8   |   18.3   |
> > > | Memory (GB) | 1.366 | 1.372 | 1.367 | 1.377 |  4.986  | 3.564 |  2.647  |  2.165   |
> > >
> > > **Table 2-9. Prediction results with 336 input length on 8 datasets.**
> > >
> > >
> > > | Exchange (MSE\|MAE) |      96      |              |     192      |              |     336      |              |     720      |              |
> > > | ------------------------ | :----------: | :----------: | :----------: | :----------: | :----------: | :----------: | :----------: | :----------: |
> > > | CrossGNN w/o RevIN       | $\underline{0.079}$ |    0.198     | $\underline{0.168}$ | $\underline{0.292}$ | $\underline{0.319}$ | $\underline{0.399}$ | $\underline{0.650}$ | $\underline{0.597}$ |
> > > | CrossGNN +RevIN          |  **0.076**   |  **0.193**   |  **0.163**   |  **0.289**   |  **0.311**   |  **0.396**   |  **0.641**   |  **0.590**   |
> > > | PatchTST w/o RevIN       |    0.089     |    0.211     |    0.179     |    0.304     |    0.341     |    0.413     |    0.792     |    0.648     |
> > > | PatchTST + RevIN         |    0.080     | $\underline{0.196}$ |    0.171     |    0.294     |    0.327     |    0.408     |    0.656     |    0.602     |
> > >
> > > | ETTh1(MSE\|MAE)    |      96      |              |     192      |              |     336      |              |     720      |              |
> > > | ------------------ | :----------: | :----------: | :----------: | :----------: | :----------: | :----------: | :----------: | :----------: |
> > > | CrossGNN w/o RevIN | $\underline{0.374}$ | $\underline{0.398}$ | $\underline{0.405}$ | $\underline{0.418}$ |  **0.410**   |  **0.415**   | $\underline{0.445}$ | $\underline{0.459}$ |
> > > | CrossGNN + RevIN   |  **0.363**   |  **0.392**   |  **0.401**   |  **0.409**   | $\underline{0.411}$ | $\underline{0.416}$ |  **0.440**   |  **0.453**   |
> > > | PatchTST w/o RevIN |    0.388     |    0.412     |    0.429     |    0.436     |    0.456     |    0.459     |    0.493     |    0.499     |
> > > | PatchTST + RevIN   |    0.375     |    0.399     |    0.414     |    0.421     |    0.431     |    0.436     |    0.449     |    0.466     |

---

> > > > ### Author Response · Authors · 2023-08-17
> > > > **Response to  follow-up questions of Reviewer F3ix (Part 2/3)**
> > > >
> > > > | ETTh2(MSE\|MAE)    |      96      |              |     192      |              |     336      |              |     720      |              |
> > > > | ------------------ | :----------: | :----------: | :----------: | :----------: | :----------: | :----------: | :----------: | :----------: |
> > > > | CrossGNN w/o RevIN | $\underline{0.266}$ | $\underline{0.333}$ | $\underline{0.336}$ | $\underline{0.375}$ |    0.355     | $\underline{0.391}$ | $\underline{0.377}$ |    0.425     |
> > > > | CrossGNN + RevIN   |  **0.259**   |  **0.329**   |  **0.319**   |  **0.371**   |  **0.322**   |  **0.380**   |  **0.371**   |  **0.417**   |
> > > > | PatchTST w/o RevIN |    0.293     |    0.347     |    0.357     |    0.389     |    0.363     |    0.395     |    0.418     |    0.438     |
> > > > | PatchTST + RevIN    |    0.274     |    0.336     |    0.339     |    0.379     | $\underline{0.331}$ |  **0.380**   |    0.379     | $\underline{0.422}$ |
> > > >
> > > > | ETTm2(MSE\|MAE)       |         96          |                     |         192         |                     |         336         |                     |         720         |                     |
> > > > | --------------------- | :-----------------: | :-----------------: | :-----------------: | :-----------------: | :-----------------: | :-----------------: | :-----------------: | :-----------------: |
> > > > | CrossGNN w/o RevIN    | $\underline{0.161}$ |      **0.252**      | $\underline{0.218}$ | $\underline{0.290}$ | $\underline{0.277}$ | $\underline{0.324}$ |      **0.360**      | $\underline{0.381}$ |
> > > > | CrossGNN + RevIN      |      **0.159**      | $\underline{0.253}$ |      **0.215**      |      **0.293**      |      **0.276**      |      **0.323**      | $\underline{0.361}$ |      **0.378**      |
> > > > | PatchTST w/o PatchTST |        0.169        |        0.260        |        0.231        |        0.301        |        0.297        |        0.345        |        0.425        |        0.431        |
> > > > | PatchTST + RevIN      |        0.165        |        0.255        |        0.220        |        0.292        |        0.278        |        0.329        |        0.367        |        0.385        |
> > > >
> > > > | ETTm1(MSE\|MAE)    |         96          |                     |         192         |                     |         336         |                     |         720         |                     |
> > > > | ------------------ | :-----------------: | :-----------------: | :-----------------: | :-----------------: | :-----------------: | :-----------------: | :-----------------: | :-----------------: |
> > > > | CrossGNN w/o RevIN |        0.294        |      **0.340**      |      **0.331**      | $\underline{0.367}$ |        0.367        | $\underline{0.389}$ | $\underline{0.418}$ | $\underline{0.420}$ |
> > > > | CrossGNN + RevIN   | $\underline{0.291}$ | $\underline{0.341}$ | $\underline{0.332}$ |      **0.364**      |      **0.363**      |      **0.385**      |      **0.414**      |      **0.417**      |
> > > > | PatchTST w/o RevIN |        0.313        |        0.364        |        0.356        |        0.389        |        0.390        |        0.415        |        0.432        |        0.437        |
> > > > | PatchTST + RevIN   |      **0.290**      |        0.342        | $\underline{0.332}$ |        0.369        | $\underline{0.366}$ |        0.392        |        0.420        |        0.424        |

---

> > > > > ### Author Response · Authors · 2023-08-17
> > > > > **Response to  follow-up questions of Reviewer F3ix (Part 3/3)**
> > > > >
> > > > > | Traffic(MSE\|MAE)  |         96          |                     |         192         |                     |         336         |                     |         720         |                     |
> > > > > | ------------------ | :-----------------: | :-----------------: | :-----------------: | :-----------------: | :-----------------: | :-----------------: | :-----------------: | :-----------------: |
> > > > > | CrossGNN w/o RevIN |        0.375        |        0.267        |      **0.381**      |        0.268        |        0.422        |        0.271        |        0.439        |        0.293        |
> > > > > | CrossGNN + RevIN   |      **0.365**      |      **0.250**      | $\underline{0.385}$ | $\underline{0.260}$ |      **0.396**      | $\underline{0.266}$ |      **0.430**      |      **0.286**      |
> > > > > | PatchTST w/o RevIN |        0.399        |        0.274        |        0.426        |        0.270        |        0.444        |        0.289        |        0.477        |        0.294        |
> > > > > | PatchTST + RevIN   | $\underline{0.367}$ | $\underline{0.251}$ | $\underline{0.385}$ |      **0.259**      | $\underline{0.398}$ |      **0.265**      | $\underline{0.434}$ | $\underline{0.287}$ |
> > > > >
> > > > >
> > > > >
> > > > >
> > > > > | Weather(MSE\|MAE)  |         96          |                     |         192         |                     |         336         |                     |         720         |                     |
> > > > > | ------------------ | :-----------------: | :-----------------: | :-----------------: | :-----------------: | :-----------------: | :-----------------: | :-----------------: | :-----------------: |
> > > > > | CrossGNN w/o RevIN | $\underline{0.150}$ |      **0.197**      | $\underline{0.196}$ | $\underline{0.241}$ |        0.252        | $\underline{0.282}$ | $\underline{0.315}$ | $\underline{0.332}$ |
> > > > > | CrossGNN + RevIN   |      **0.148**      | $\underline{0.200}$ |      **0.195**      |      **0.240**      |      **0.240**      |      **0.281**      |      **0.311**      |      **0.329**      |
> > > > > | PatchTST w/o RevIN |        0.154        |        0.212        |        0.206        |        0.251        |        0.255        |        0.287        |        0.334        |        0.347        |
> > > > > | PatchTST + RevIN   |        0.152        |        0.199        |        0.197        |        0.243        | $\underline{0.249}$ |        0.283        |        0.320        |        0.335        |
> > > > >
> > > > > | Electricity(MSE\|MAE) |      96      |              |     192      |              |     336      |              |     720      |              |
> > > > > | --------------------- | :----------: | :----------: | :----------: | :----------: | :----------: | :----------: | :----------: | :----------: |
> > > > > | CrossGNN w/o RevIN    |    0.134     | $\underline{0.221}$ |  **0.144**   | $\underline{0.238}$ |    0.174     | $\underline{0.260}$ | $\underline{0.198}$ | $\underline{0.290}$ |
> > > > > | CrossGNN + RevIN      |  **0.129**   |  **0.219**   | $\underline{0.145}$ |  **0.235**   | $\underline{0.170}$ |  **0.259**   |  **0.193**   |  **0.289**   |
> > > > > | PatchTST w/o RevIN    |    0.139     |    0.237     |    0.149     |    0.245     |    0.173     |    0.265     |    0.201     |    0.295     |
> > > > > | PatchTST + RevIN      | $\underline{0.130}$ |    0.222     |    0.148     |    0.240     |  **0.167**   |    0.261     |    0.202     |    0.291     |
> > > > >
> > > > >
> > > > >
> > > > > In brief, when not using RevIN, CrossGNN can still outperform PatchTST with RevIN in prediction effectiveness with the same input length, meanwhile equipping a more lightweight model design. **Finally, we can conclude that our proposed CrossGNN sheds light on significant potential of GNNs in modeling cross-time (temporal) and cross-variable  (variable-level) interactions for long-term MTS prediction.**
> > > > >
> > > > >
> > > > >
> > > > > Thank you for your follow-up questions. We believe this additional results and explanations can help readers better understand our work, and we will carefully involve these materials into our main texts or Appendix. Thanks again!
> > > > >
> > > > > Authors of Paper 3421

---

> > > > > > ### Comment · Reviewer_F3ix · 2023-08-18
> > > > > >
> > > > > > Thank you very much for the results. Base on these results and comments from other reviewers on the novelty and performance, I changed my score to 5.

---

> > > > > > > ### Author Response · Authors · 2023-08-19
> > > > > > > **Thanks for positive follow-up**
> > > > > > >
> > > > > > > Dear Reviewer F3ix,
> > > > > > >
> > > > > > > We gratefully appreciate your positive follow-up. Your comments will encourage us to keep improving the quality of our manuscript to enable it satisfy the high-quality requirement of NeurIPS 2023. Thanks very much!
> > > > > > >
> > > > > > >
> > > > > > > Authors of Paper 3421

---

### Author Rebuttal · Authors · 2023-08-09

Dear Reviewers,

We would like to thank you for your valuable time and constructive comments on our manuscript and we have made sufficient improvements of our work according to your comments. Here we list the major improvements below.

- **Enhanced Clarifications**: We have enhanced our manuscript with more in-depth explanations where necessary, such as an explanation of ablation studies.

- **Additional Experiments**: Based on the feedback, we have conducted additional experiments to further validate the effectiveness of our proposed model. These include but not limited a comparison with recent SoTA method such as RLinear, PatchTST, Crossformer, a comparison of running time  among well-acknowledged deep forecasting models.

- **Clearer Definitions**: We have added clear definitions for terms like 'heterogeneity between variables' to ensure readers have a comprehensive understanding of our method.


**Technical novelty of CrossGNN.**

The contribution of CrossGNN extends highlighting the challenges of unexpected noise and variable-wise heterogeneity. It also introduces significant innovations in enhancing interactions across time and variable dimensions. Next, we will elaborate our technical novelty and  highlight the distinctions between CrossGNN and other SoTA methods.

- **In time interaction.**

  1. **First attempt to  explore GNN for temporal modeling.**

    To the best of our knowledge, this is the first attempt to explore GNN for capturing temporal  relationships, which is beyond capturing variable-level correlation in  previous literature. We elucidate how to exploit GNN to connect different temporal points to  learn a stable and general temporal graph, which can be  free from noise interference.

  2. **First attempt to propose Cross-Scale interaction.**

  We  emphasize the significance of point-level temporal interactions across scales.  Based on the insight that coarse temporal scale extracts the backbone of regularity, we introduce the cross-scale module, which directly extracts  coarser-scale information and extends the temporal learning towards multiple scales. CrossGNN allows  refined interaction learning across different temporal scales, resulting in noise-reduced effects.

  3. **Adaptable multi-scale MTS acquisition.**

    We devised an Adaptive Multi-Scale Identifier (AMSI), capable of dynamically partitioning scales based on input sequence frequency characteristics. This distinguishes our approach from prior methods that rely on fixed-length manual scaling, contributing to an automatic multi-scale sequence acquisition.

- **In variable interaction.**

  1. **Pioneering temporal data heterogeneity.**

     We first  propose Cross-Variable learning via decoupling temporal homogeneous-heterogeneous relationships.

  2. **Low computation complexity.**

     We maintain the computational complexity of both the Cross-scale and Cross-variable GNN modules at a linear level by constraining the number of inactive edges.

In summary, we outline the distinctions and innovations that set us apart from the similar methods the reviewer mentioned:

- **Differences from TimesNet.**  Even TimesNet  extends 1-D  series data to a 2-D space, it still lacks explicit modeling interactions across different  temporal scales.

- **Differences from Crossformer.** Crossformer solely focuses on homogenous associations and ignores heterogeneity in the real world. Besides, the time complexity of Crossformer is $O(rDL)$ ($r$ for router vectors, $D$ for variable count, $L$ for sequence length), while CrossGNN is $O(D)$.

- **Differences from GNN-based Approaches.** GNN-based time series prediction methods merely  capture cross-variable relationships. However, they overlook (1) heterogeneity among variables, (2) complexity of $O(N^2)$, without graph pruning for efficiency, (3) the potential of GNN-based approaches in capturing temporal relationships, fail to exploit GNN to model series along temporal dimensions.

We will polish the  technical novelty in our manuscript based on above clarification.


Finally, we believe that these revisions have significantly improved our manuscript. We hope that our responses and the changes made address the concerns of the reviewers adequately.

Once again, thank you for your time and effort in reviewing our work. We look forward to your continued feedback.

Best regards,

Authors of Paper 3421

---

> ### Author Response · Authors · 2023-08-21
> **Thanks for Area Chair and Reviewers!**
>
> Dear Area Chair and Reviewers,
>
> Thanks for your assistance and constructive suggestions on our work.
>
> We believe our work contributes to a  lightweight time series learning framework to confront temporal fluctuation and noise, experiencing an  $O(L)$  spatial-temporal complexity. The main technical contributions lie in  **1) first explore GNN to model temporal dependence, 2)  advance cross-scale temporal interactions by identifying and interacting multi-scale temporal regularity, 3) first attempt to decouple the heterogeneous and homogeneous correlations within MTS thus enabling a more stable learning process.**
>
> **We gratefully acknowledge the achieved consistency of the positive evaluations from all reviewers.** Currently, we have incorporated new experiments and discussions based on the valuable suggestions of reviewers into the ongoing polishment of our manuscript. We hope that the reviewers can continue to support our work in the subsequent stages.
>
>
> Thanks again for your valuable time!
>
> Yours sincerely,
>
> Authors of Paper 3421

---

### Decision · Program_Chairs · 2023-09-21

**Decision:**

Accept (poster)

**Comment:**

The authors propose a GNN-based method for time series modeling to captures  temporal fluctuations and heterogeneity between different variables and to model multi-scale time series. The reviewers praised the clarity of the presentation, the efficiency of the method, its comprehensive ablation studies and high performance, especially considering the low time complexity. During the discussion phase, the authors addressed the main concerns raised by the reviewers. Following the discussion, the reviewers unanimously accepted the paper. Due to the importance of time series analysis, the community would benefit from the publication of the paper and the availability of the code for this method, so I recommend acceptance.